# Continentality determines warming or cooling impact of heavy rainfall events on permafrost

Alexandra Hamm [1,2] ✉, Rúna Í. Magnússon [3], Ahmad Jan Khattak[4] & Andrew Frampton [1,2]

Permafrost thaw can cause an intensification of climate change through the release of carbon as greenhouse gases. While the effect of air temperature on permafrost thaw is well quantified, the effect of rainfall is highly variable and not well understood. Here, we provide a literature review of studies reporting on effects of rainfall on ground temperatures in permafrost environments and use a numerical model to explore the underlying physical mechanisms under different climatic conditions. Both the evaluated body of literature and the model simulations indicate that continental climates are likely to show a warming of the subsoil and hence increased end of season active layer thickness, while maritime climates tend to respond with a slight cooling effect. This suggests that dry regions with warm summers are prone to more rapid permafrost degradation under increased occurrences of heavy rainfall events in the future, which can potentially accelerate the permafrost carbon feedback.

Permafrost soils store approximately 1600 Pg of organic carbon, which is twice the amount of carbon currently in the earth's atmosphere[1–3]. While immobilized at present, this carbon reservoir can be decomposed and released as greenhouse gases, such as carbon dioxide or methane when thawed and further enhance global warming[4]. Permafrost soils in Arctic and alpine regions have responded to global warming with an increase in thickness of the seasonally thawed layer overlaying the permafrost, the active layer[5–7]. The thickness of the active layer is crucial for our understanding of the amount of carbon available for decomposition and release as greenhouse gases and affects soil depth, hydrology, and disturbance dynamics of permafrost ecosystems[8–11]. Furthermore, changes in active layer thickness can cause changes to the hydrological connectivity of the landscape[9], to ecosystems and ecosystem services[12], as well as damage to infrastructure built on permanently frozen soils through soil subsidence and thermokarst[13,14].

While traditionally air temperature and snow cover are considered to be the main climatic factors to influence active layer depth, emerging evidence suggests an important role of summer precipitation[7,15–17]. The role of air temperature and of snow cover have been studied widely for decades and have been subject to systematic review[18,19]. However, no synthesis on the role of summer precipitation in active layer thermal dynamics is available, and a clear perspective of the role of summer precipitation in future permafrost degradation and associated greenhouse gas emissions is lacking[7].

Precipitation in the Arctic is expected to increase. Current estimates predict local increases of up to 40% (e.g., in the Canadian Arctic and Eastern Siberia) over the 21st century[20,21] with a strong increase in summer due to poleward moisture transport[21] and the increased proportion of precipitation falling as rain rather than snow[22]. Furthermore, part of this increased summer precipitation will occur in the form of extreme precipitation events[23–25]. This is an important factor in permafrost landscapes as rainfall changes the soil hydrothermal properties due to changes in water content. The ability to conduct heat as well as to store energy greatly depends on soil moisture and influences the overall energy directed into the ground and thereby affects permafrost degradation[26]. Furthermore, rainfall during warm summer days may also contribute to advective heat transport into the soil, which directs

[1]Department of Physical Geography, Stockholm University, Stockholm, Sweden. [2]Bolin Centre for Climate Research, Stockholm University, Stockholm, Sweden. [3]Plant Ecology and Nature Conservation Group, Wageningen University, Wageningen, Netherlands. [4]NOAA Affiliate at Lynker, Office of Water Prediction, National Water Center, Tuscaloosa, AL, USA. ✉e-mail: alexandra.hamm@natgeo.su.se

more energy from the atmosphere through the rainwater towards the permafrost[15,27]. Hence, anticipated future increases in rainfall and heavy rainfall events could affect permafrost soils and regulate their response to warming air temperatures.

Opposing effects of heavy summer rainfall on ground temperatures have been reported in the literature, often hinting at rain-induced changes in the hydrothermal soil properties (such as thermal conductivity and heat capacity) or changes to surface heat fluxes and the surface energy balance. Additionally, opposing effects are not only found between different sites, but also between different observation depths within the soil profile. Here, we hypothesize that the effect varies with prevailing climatic conditions and collect relevant literature published on the effect of heavy summer rainfall on the active layer throughout different permafrost landscapes ranging from high Arctic over sub Arctic to Alpine permafrost regions. In order to understand the role of increasing rainfall and heavy rainfall events, we summarize and synthesize those findings, and categorize the effects into distinct processes that cause soils to react differently to heavy summer rainfall. Based on this analysis, we proceed to investigate the governing physical mechanisms causing the different ground temperature responses using a physics-based numerical model.

## Results

### Literature review

We screened literature found through systematic database searching, snowball searching and Google Scholar alerts for eligibility using PICOS criteria and a PRISMA workflow (Supplementary Text 1.1). This way, we identified 22 peer-reviewed articles that address the impact of heavy summer rainfall on permafrost soil temperatures and found considerable variability in reported effects on the soil thermal regime. The studies show a wide spatial distribution across permafrost ecosystems, including Alaska, Greenland, Siberia and the Tibetan Plateau (Fig. 1). The identified papers were published between 2001 and 2022 and cover a range of different analysis methods. Effects of heavy rainfall on permafrost has seen increasing attention in scientific literature in recent years, with increasingly more studies using a combined field- and modeling approach to investigate the effect. A detailed overview of the identified studies is presented in Table S1.

We used a vote-counting approach with a pre-defined rule set to assign identified studies to different response classes (Supplementary Text 1.2), based on the reported net effect of heavy summer rainfall (modeled or observational) on overall subsoil temperatures. We assigned studies to "cooling" for reduced net soil temperatures, for reduced thaw depth, or reduced active layer thickness (ALT), or "warming" for higher net soil temperatures, or increased ALT or thaw depth, in response to heavy summer rainfall. In some cases, studies showed variable effects ("both") depending on the time frame that the study has encompassed (one summer up to several decades), the intensity of heavy summer rainfall, or the type of manipulation experiment (e.g., heavy rainfall scenarios or gradual increase in precipitation). If divergent effects at different depths are reported, the effect is further distinguished into topsoil and subsoil warming or cooling (here, topsoil: ≤10 cm and subsoil: >10 cm, see Table S2). As soil temperatures close to the thaw front (subsoil) are indicative of thaw depth development we further classify a study as "warming", if subsoil warming is observed and "cooling" if the subsoil experiences cooling. Furthermore, one study did not see a temperature response to heavy summer rainfall ("no effect")[28]. We also consider two studies that address the effect of wetter soil conditions through soil moisture, which is not necessarily always coupled to precipitation[29,30]. However, the results are relevant for the general understanding of soil moisture dynamics and active layer temperature response.

With this categorization, we attributed the "cooling" effect to seven studies[31–37], "warming" was observed in twelve studies[15–17,27,29,38–44], two studies showed both effects[30,45] depending on

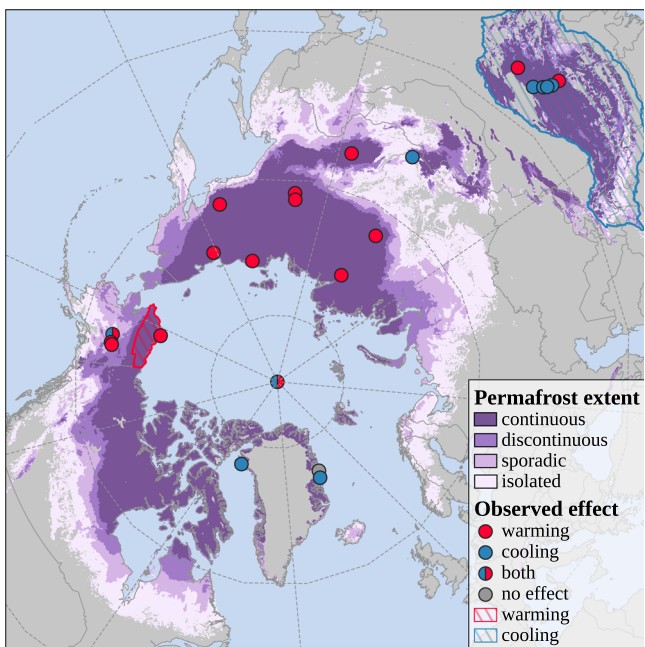

**Fig. 1 | Location of study sites included in the literature review.** Shades of purple indicate the permafrost extent (continuous (dark) to sporadic (light)[80]). Red circles indicate that an overall ground warming effect was noted in the study, blue circles indicate a cooling effect. Blue and red circles indicate that both effects were reported and gray circles that no effect was observed. The red and blue circle right on the north pole indicates that the corresponding study uses data from all over the Northern Hemisphere[45]. Red and blue hashed areas indicate larger scale modeling studies reporting overall warming in Northslope, Alaska[16], and predominantly cooling over the Qinghai-Tibet Plateau[35], respectively. Basemap data was retrieved from https://thematicmapping.org/. Source data are provided as a Source Data file.

seasonal timing or rainfall event magnitude, and one study did not see a temperature response to heavy summer rainfall[28]. Where available, the magnitude of the warming and/or cooling effect is quantified and noted in Table S2. In the subsoil, the temperature effect ranges from 0.06 °C to 1 °C warming (quantified in seven out of 22 studies, see Table S2). In the topsoil, cooling of up to −10 °C was observed (quantified in seven out of 22 studies, see Table S2). In general, studies that report thermal effects over multiple soil depths find warming of subsoils and cooling of the topsoil (see Table S2). Not all studies report temperature response but rather report changes in thaw depth or ALT[15,29,30,38] or surface heat fluxes[42] to describe the effect of heavy summer rainfall. The literature findings indicate that reported effects can already vary based on experimental and observational criteria such as the duration and depth of monitoring of soil temperatures (Table S1).

To reduce methodological differences among studies, we further narrowed down our selection to studies that report field-measured changes in active layer thickness, thaw depths or soil temperatures in spatially explicit locations in the same summer as the heavy rainfall event or experimental treatment (see Supplementary Text 1.2). For the remaining 14 study sites, we find that sites for which warming effects of heavy rainfall were reported are generally situated in more continental regions compared to sites for which cooling effects were reported. From the 14 selected study sites, we analyzed site specific climatological data using the location reported in the studies (seven reporting subsoil warming, six reporting subsoil cooling, and one study reporting no effect) and ERA5 reanalysis data as well as the Conrad's Continentality Index (Table S3). To identify climatological contrasts that may explain variability in the soil thermal response across the current body of evidence, we performed a principal component analysis (PCA). The result indicates that the climatology of these sites mainly varied in

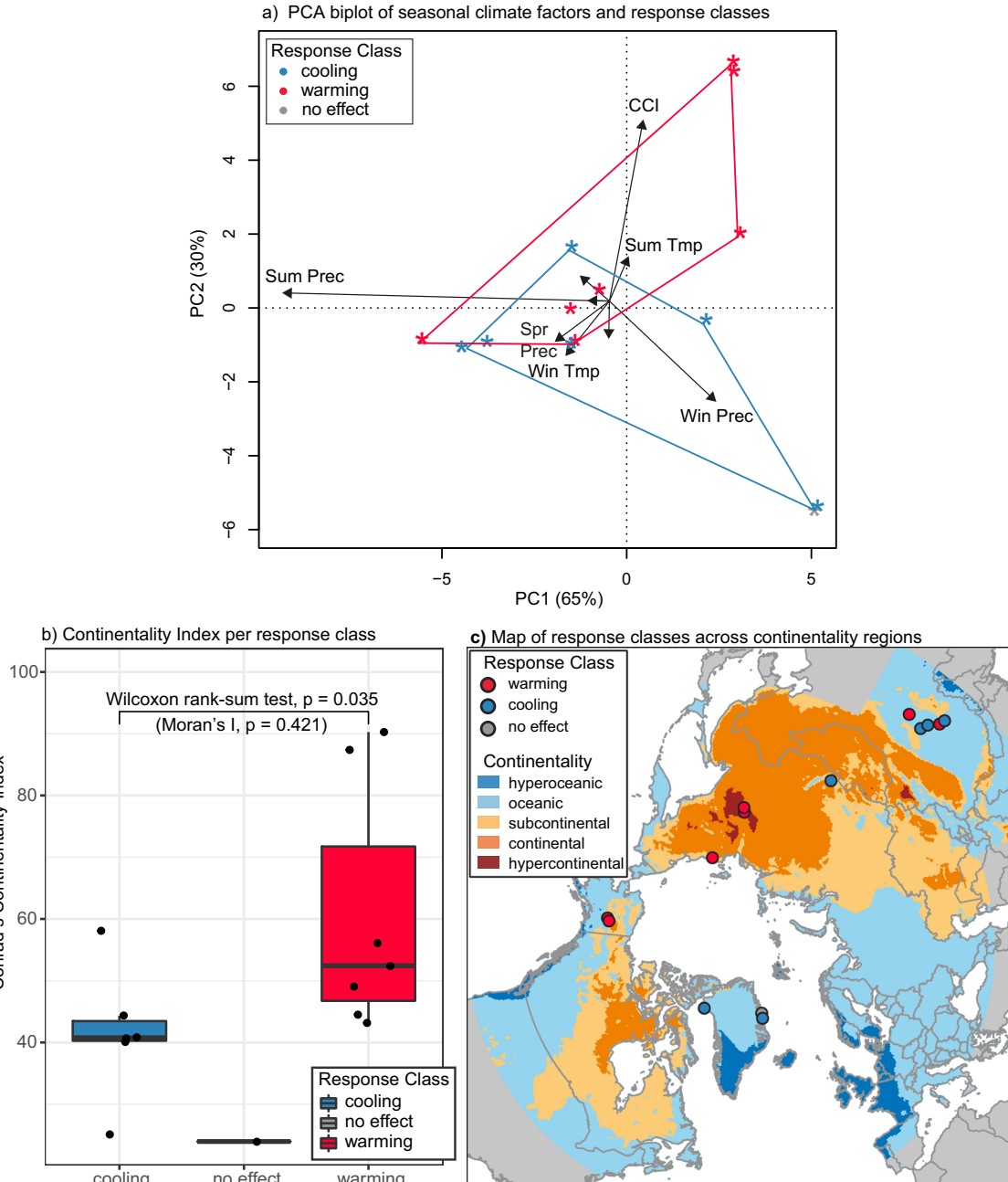

**Fig. 2 | Climatological contrasts among sites for which field-measured effects of heavy rainfall on the soil thermal regime are reported based on monitoring or experimental studies. a** Principal Component Analysis (PCA) biplot of seasonal averaged ERA5 climatological variables for selected sites, where arrows represent principle component (PC) loadings of averaged temperature (Temp) and precipitation (Prec) in spring (Spr), summer (Sum), fall (Fall, labels not shown here due to low PC loadings), and winter (Win) as well as the Conrad's Continentality Index (CCI) for selected sites (Table S3). Points represent the individual sites, colored by reported effect of heavy rainfall on the soil thermal regime. **b** Boxplot of CCI of selected sites, grouped and colored by warming or cooling as reported effect. Center lines represent the median, box limits represent upper and lower quartiles, whiskers represent 1.5 times the interquartile range and points represent individual data points. Moran's I indicates no significant spatial autocorrelation among residuals. **c** Map of reported effects from selected sites over map of Conrad's Continentality Index, with hyperoceanic (dark blue, CCI < 20), oceanic (blue, CCI < 50), subcontinental (yellow, CCI < 60), continental (orange, CCI < 80), and hypercontinental (dark orange, CCI > 80) classifications. Red, blue, and gray circles indicate the observed effect in each study used for the satistical analysis (red: warming, blue: cooling, gray: no effect) All meteorological data used in (**a**) and used to calculate CCI in (**b**) and (**c**) is based on ERA5 reanalysis for the time span 1991–2020. Basemap data in (**c**) was retrieved from https://thematicmapping.org/. Source data are provided as a Source Data file.

terms of continentality and total summer and winter precipitation (Fig. 2a, black arrows). We observed a pattern of higher continentality (and associated higher summer temperature and lower winter temperature) and lower winter precipitation in sites that showed warming effects of heavy rainfall events, as opposed to lower continentality and higher winter precipitation for sites that showed cooling effects (Fig. 2a). Wilcoxon rank sum tests confirm that sites for which warming effects are reported have significantly higher continentality indices ($p = 0.035$, $n = 14$, Fig. 2b, c) but indicated no significant difference in winter precipitation ($p = 0.731$, $n = 14$) compared to sites that showed cooling effects of heavy rainfall events. Although the current body of literature is limited and variable in methodology and rainfall event magnitudes, this suggests that warming effects of heavy rainfall events may mostly be observed in sites with continental climates and,

accordingly, warmer summer and colder winters, whereas cooling effects may be observed in sites with maritime climates.

Most studies discuss potential driving mechanisms of observed cooling or warming in response to heavy rainfall events (see Table S1). Generally, they can be summarized in two groups; hydrothermal properties of the soil, and surface heat fluxes. Hydrothermal soil properties such as thermal conductivity and heat capacity are affected by the available soil moisture, which is increased in wet summers. While thermal conductivity increases in wet soils and leads to higher heat conduction from the atmosphere into the subsurface, an increase in heat capacity (including latent heat) enhances energy storage but also causes slower thaw. Surface heat fluxes such as heat advection from rain (entering the ground as energy gain) and evaporation (leaving the ground as energy loss) can cause the soil to warm and cool, respectively, within a single summer. In multi-year analysis, increased latent heat requirements of wetter soils can add to delayed thaw, shallower thaw depths, and overall colder temperatures in the following year. On the other hand, increases in thermal conductivity may persist over multiple years and cause faster warming and deeper active layers. Despite all those mechanisms theoretically being active in all permafrost landscapes and climatological settings similarly, the observed effects on ground temperatures in the studies investigated here can show opposing effects in the temperature response to increased summer rainfall.

## Modeling the effect of heavy summer rainfall on permafrost

Based on findings from the literature review and the apparent contrast in ground warming or cooling based on a continentality gradient, we designed a model study to investigate the different processes causing the observed temperature responses. We use a state-of-the-art physics-based numerical model (Advanced Terrestrial Simulator, ATS v1.2[46], see Methods section and Supplementary Text 2) and created four contrasting and complementary climate scenarios, representative of the observed range of climatological settings from the synthesized literature: a (1) cold and wet, (2) cold and dry, (3), warm and wet, and (4) warm and dry summer climate scenario (see Fig. S1 for representation of weather data). The warm climate is characterized by a comparably warm mean annual air temperature (MAAT) and a high annual temperature amplitude (warm summers, cold winters). The cold climate has a lower MAAT and low annual temperature amplitude (mild summers, mild winters). The wet climate is characterized by a high summer precipitation amount, while the dry climate has less summer rainfall (see Table 1 for full summary). These scenarios were developed to represent the range of climatological conditions in sites from the synthesized literature and can be conceptualized as end members along a gradient from a continental climate (warm-dry summer) to a maritime climate (cold-wet summer). The warm and dry scenario with cold winters is representative of most highly continental regions with high annual temperature amplitudes, such as the eastern Siberia permafrost region. The cold and wet scenario with lower annual temperature amplitudes and rainier summers is generally representative of coastal Greenland and Alaska, as well as the Tibetan Plateau (see Fig. 2c). To simulate heavy summer rainfall, we change summer precipitation by increasing the rainfall of one day on the 15th of each summer month (June, July, August). The resulting difference in soil temperature over time between the reference case without heavy rain events (ref. case) and the heavy rain case (HR case) constitutes the main variable of analysis.

We find that the four climate scenarios yield distinct opposing effects in the soil temperature response under heavy summer rainfall conditions. The temperature difference is described as HR case temperature minus ref. case temperature in the topsoil (25% of the maximum ALT) and the subsoil (75% of the maximum ALT) on a daily time-step during the thawing season of the same year in which heavy rainfall was applied (for absolute temperatures see Fig. S2). While both dry climates (warm-dry and cold-dry) show a warming of the subsoil following heavy summer rainfall of up to 0.3 °C and 0.2 °C, respectively (Fig. 3a, yellow lines), both wet climates respond with an initial warming, followed by a cooling effect. However, the cooling effect seen in the wet climates (up to −0.12 °C and −0.08 °C for the cold-wet and warm-wet climate, respectively, Fig. 3a, blue lines) is comparatively not as strong as the warming effects in the dry climates (Fig. 3a, yellow lines).

While subsoil temperature responses show effects in opposite directions, topsoil temperature responses do not follow the same effect pattern. In all climates, except warm-dry, a heavy rain event in early summer causes the top layers of the soil to be colder than under average conditions in the reference case (up to −0.4 °C, Fig. 3b). Afterwards, the soil in the HR cases approaches similar conditions as the reference case again until the next rain event. In the warm-dry climate, an overall warming gets intersected by cooling effects after the heavy rain events on July and August 15. The simulated magnitude of temperature responses is in line with observed temperature effects in the literature, which show that the subsoil temperatures in most studies tend to increase, while topsoil temperatures tend to decrease (Table S2).

Soil temperatures close to the thawing front (i.e., in the subsoil) are indicative of the ALT development. Active layer thaw is overall fastest in the HR case in the warm-dry climate scenario (Fig. 3c, dashed yellow line and shaded area), while the HR case cold-wet climate scenario experiences the slowest thaw (Fig. 3c, solid blue line and shaded area). Maximum active layer thickness in the wet climates exhibits no difference between the reference and HR case (cold-wet: −0.49 m, warm-wet: −0.67 m in the ref. case as well as the HR case). However, the dry climate simulations show an increase of 2 cm in maximum ALT from −0.69 m to −0.71 m in the warm-dry simulation and from −0.53 m to −0.55 m in the cold-dry simulation and significant delay in autumn freeze-up. Hence, our model results demonstrate that heavy rainfall can affect ALT and associated permafrost degradation processes differently in wet and dry climates, such that greater changes should be expected in dry climates.

## Driving physical mechanisms

Our results from literature review and model analysis showed differential responses of permafrost soil temperatures in different climatological contexts, and with soil depth (i.e., a tendency for cooling of topsoils and warming of subsoils). Using the model simulations for the generic climate scenarios, we assessed the relative contribution of different physical mechanisms affecting the soil thermal regime. One of the most relevant processes affecting soil temperatures in permafrost landscapes to help explain these observations is vertical heat conduction[47]. Heat conduction depends on thermal conductivity of the material, which is generally higher in wet soils than in dry soils[48]. The relative increase of thermal conductivity, however, is higher in dry soils than in wet soils. While wet soils are already highly conductive, dry soils gain a strong increase in thermal conductivity due to increased saturation caused by heavy rainfall. In our simulations, bulk thermal conductivity in the topsoil, which acts as the contact layer for energy transfer from the atmosphere into the soil, increases by almost 35% in the dry climate scenarios, but only by up to 17% in the wet climate scenarios (Fig. 4 and Fig. S3a). This contributes to increased heat conduction from the atmosphere also to deeper soil layers (Fig. S4 and Fig. S5). In the case of the dry climate scenarios, this increase is substantial enough to increase subsoil temperatures and even ALT by up to 2 cm. Additionally, due to faster warming in spring and summer, air temperature warming rates are higher in the continental climate scenarios causing greater gradients between air and surface and thereby delivering more heat from the atmosphere to the soil. Heat advection through rainfall was noticeable in all cases, but overall small when compared to heat conduction (Fig. S6).

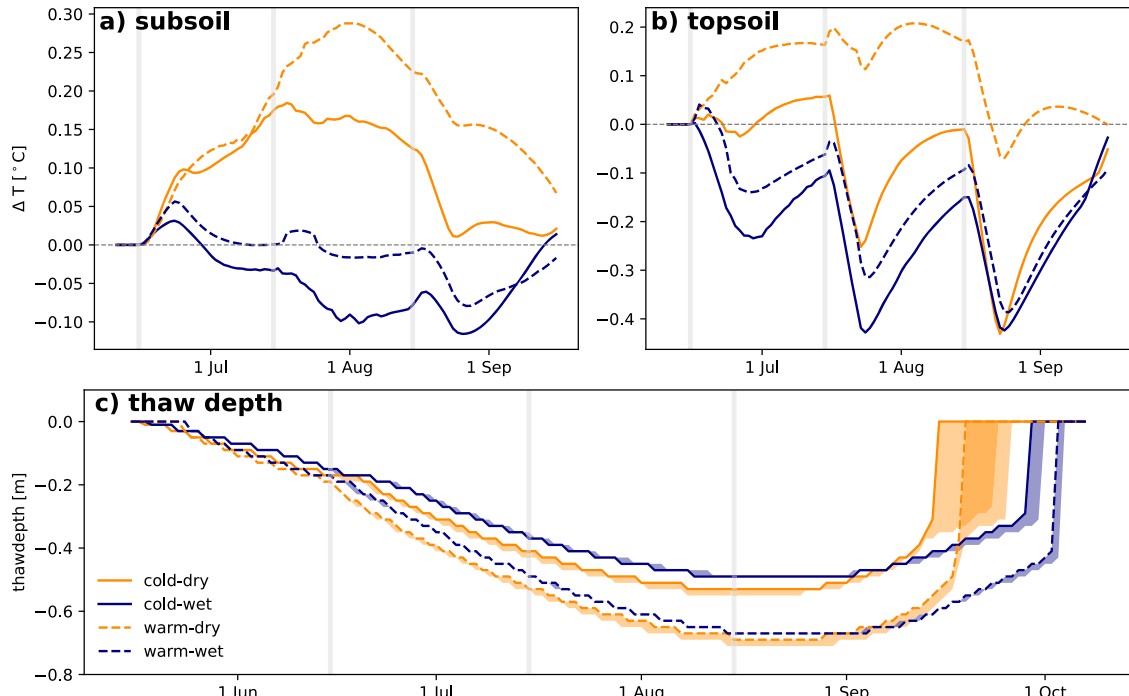

**Fig. 3 | Effect of heavy summer rainfall on ground temperatures and active layer development.** Temperature differences between heavy rain (HR) and reference case are displayed from the first heavy rain event until the end of the thawing season in (**a**) the subsoil (75% of maximum active layer thickness) and (**b**) the topsoil (25% of maximum active layer thickness). Differences are displayed as HR case minus reference case. The dry climate scenarios are represented by yellow lines, the wet scenarios by blue lines. Dashed and solid lines indicate the difference between cold (solid) and warm (dashed) climate scenarios. Daily values are smoothed over a 7-day window. Gray vertical bars indicate the timing of the simulated heavy rain events in each summer month (June, July, August). Daily modeled thaw depth (**c**) is represented as the 0 °C isotherm depth in each scenario. Solid and dashed lines represent the reference case in each climate scenario and the upper/lower edged of the shaded areas display the maximum thaw depth of the HR case. Source data are provided as a Source Data file.

While thermal conductivity increases energy transfer from the atmosphere into the soil in the warm season, heat capacity changes can counteract this gain. Like thermal conductivity, heat capacity increases with soil moisture and determines how fast the soil warms. Heat capacity increases in all climate scenarios as a response to rainfall (Fig. 5). However, towards the late summer season, it increases more strongly in the subsoil in the wet scenarios (by up to 12%, Fig. 5b and d and Fig. S7a, blue lines) when compared to the dry climate scenarios (up to 4%, Fig. 5a, c and Fig. S7a, yellow lines), which causes deeper layers to have an even lower warming rate than the dry scenarios. The increase in heat capacity into deeper layers in the wet scenarios can be explained by the soil moisture retention characteristics allowing water to infiltrate into deeper layers and increasing liquid saturation at a greater rate in the wet cases than in the dry cases (Figs. S8 and S9). Increased heat capacity in the active layer also leads to the observed delay in the onset of freezing (see Fig. 3c). In the topsoil, heat capacity shows a stronger increase in the dry climates (up to 45%, Fig. S7b, yellow lines) as a direct response to rainfall, while the wet climates only experience an increase of up to 25% (Fig. S7b, blue lines), contributing to the overall cooling in the topsoil in most cases as seen in Fig. 3b.

## Discussion

Both the current knowledge base and our generic model study show that continental permafrost landscapes may experience accelerated permafrost degradation through a combination of warmer air temperatures and enhanced summer precipitation. In areas characterized by a maritime climate, or more generally by a low annual air temperature amplitude and high precipitation, summer rainfall can attenuate effects of warming air temperature on permafrost degradation. A substantial part of the continental climate in the northern hemisphere permafrost landscape coincides with the location of carbon- (ref. 49, Fig. S10) and ice rich (ref. 50, Fig. S11) permafrost and

Yedoma deposits (ref. 51, see Fig. S11), which are particularly vulnerable to climate change and abrupt thaw and can enhance the permafrost carbon feedback rapidly[3]. Accelerated degradation through a combination of air temperature warming and increased precipitation in these regions can therefore have substantial consequences for future feedbacks between permafrost thaw and climate change, but also for Arctic infrastructure[13] or thermokarst development[14].

Our results (Tables S1, S2 and Fig. 3) indicate an influence of depth of observation or changes to the soil thermal regime, with cooling predominating in shallow soil layers even when thaw depths or deeper soil temperatures indicate warming. Hence, studies reporting effects of heavy rainfall events on the soil thermal regime may show limited comparability across observation methods (e.g., active layer thickness, instantaneous thaw depths, soil temperatures or heat fluxes) and observation depths. While our evaluation of reported rainfall effects against site climatology (Fig. 2) showed a balanced distribution of experimental (e.g., irrigation) and observational studies across sites where warming and cooling effects were reported (Table S1), a relatively higher proportion of studies reporting shallower soil temperature changes among sites that report cooling (Table S2) may have contributed to observed patterns. However, our model simulations independently show a similar pattern of subsoil warming and active layer deepening under climatic conditions representative of continental sites, versus cooling and no effects on active layer thickness in climates representative for maritime regions.

Utilizing a physics-based numerical model, we identified the physical mechanisms causing the observed opposing effects. Through a combination of the effect of summer rainfall on heat capacity and thermal conductivity, dry soils tend to respond with warming, while wet soils experience cooling, especially in later summer, as compared to average rainfall conditions. Other than the effects of moisture on soil heat conduction, heat advection is often brought up as an

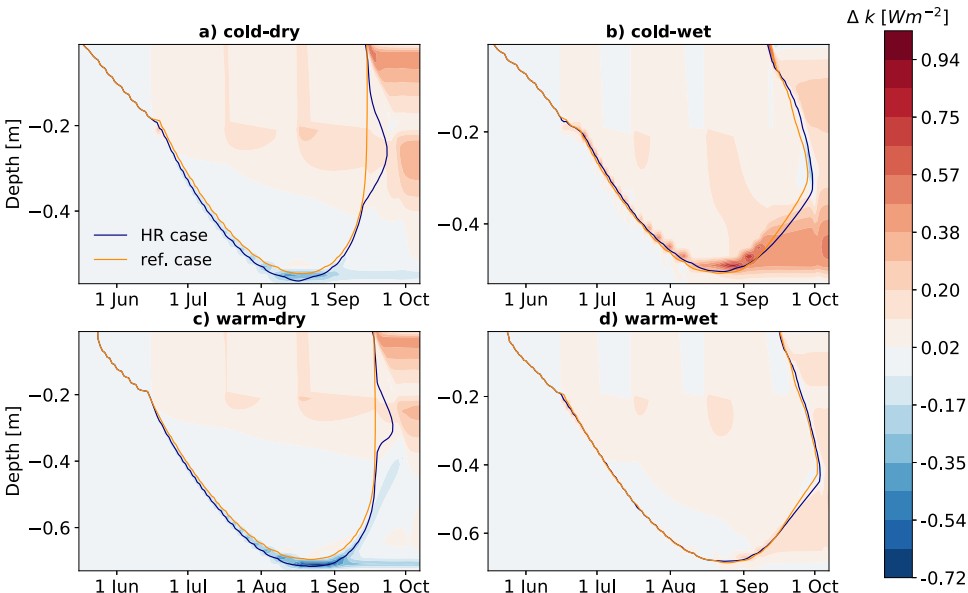

**Fig. 4 | Effect of heavy summer rainfall on soil thermal conductivity.** Daily differences in thermal conductivity ($k$) between the reference case and the heavy rain (HR) case (HR−ref. case, $\Delta k$) throughout the depth profiles in (**a**) the cold-dry, (**b**) the cold-wet, (**c**) the warm-dry, and (**d**) the warm-wet climate scenario. Shades of red indicate higher thermal conductivity in the HR case compared to the ref case, blue shades indicate lower values. The orange and blue contour represent the 0 °C isotherm depth in the reference and HR case, respectively. Differences are presented in W m⁻². Source data are provided as a Source Data file.

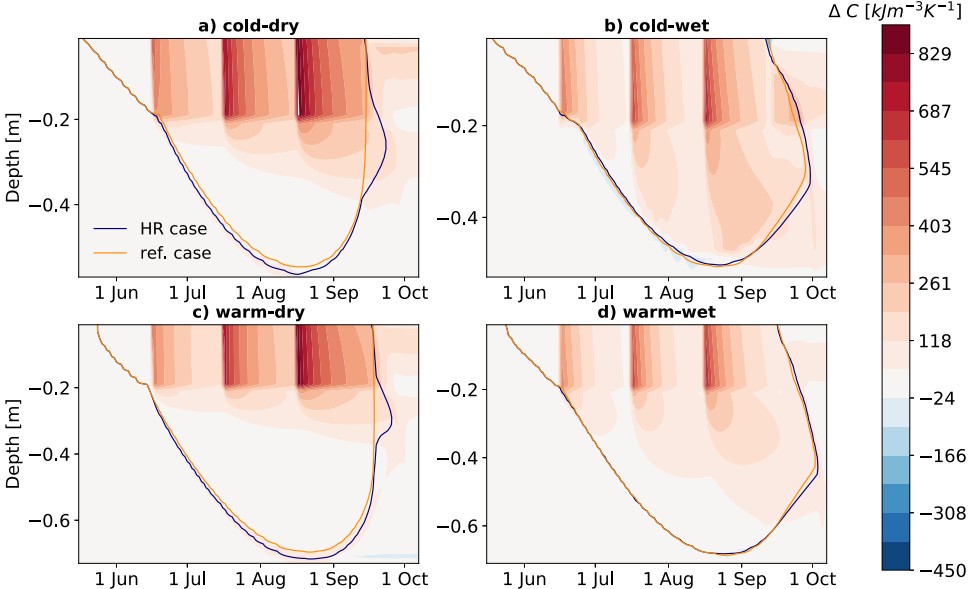

**Fig. 5 | Effect of heavy summer rainfall on soil heat capacity.** Daily differences in heat capacity ($C$) between the reference case and the heavy rain (HR) case (HR−ref. case, $\Delta C$) throughout the depth profiles in (**a**) the cold-dry, (**b**) the cold-wet, (**c**) the warm-dry, and (**d**) the warm-wet climate scenario. Shades of red indicate higher heat capacity in the HR case compared to the ref case, blue shades indicate lower values. The orange and blue contour represent the 0 °C isotherm depth in the reference and HR case, respectively. Differences are presented in kJ m⁻³ K⁻¹ Source data are provided as a Source Data file.

important factor for the ground thermal regime after summer rainfall[27]. In our simulations, vertical heat advection even during heavy rain events was small compared to heat conduction (Fig. S6), suggesting that the effect plays a small role in these simulated conditions. It is noted that heat advection also depends on the overall weather conditions, which are usually cloudy with little solar radiation during heavy rain events causing the net heat advection to be small[52] and heat conduction to be the predominant heat transport mode[53]. The role of lateral heat advection as a response to summer rainfall is still unclear, but may be of particular importance in permafrost landscapes with distinct topographical features like bogs or along slopes[27,47,54].

The identified mechanisms causing the soil thermal response in this study are based on a single representative set of soil parameters and generalized climatic conditions. We assessed the model sensitivity to climatic and non-climatic factors by including additional analysis with different magnitudes and temporal distribution of rainfall events as well as different soil compositions.

Firstly, we investigate the sensitivity of magnitude and temporal distribution of increased summer rainfall by creating three additional scenarios. One with a uniform increase of overall summer rainfall by 50% to simulate for a general increase in summer precipitation as predicted by ref. 21 (see Supplementary Text 3.1), and two scenarios

accounting for even stronger rain events (50% stronger HR case) and less intense rain events (50% less intense HR case, see Supplementary Text 3.2). We find that generally under all tested conditions, the effect direction is unchanged, while the effect extent varies from up to 0.1 °C warming in the dry cases and −0.15 °C cooling in the wet cases for the uniform rainfall distribution (Fig. S12), to a warming of more than 0.3 °C in the dry climate cases and a similar cooling of −0.15 °C in the wet cases for the 50% more intense heavy rainfall event simulation (Fig. S13). This suggests that independent of the magnitude and temporal distribution, increased summer precipitation in the form of uniformly increased summer rainfall as well as the occurrence of heavy rainfall events will generally warm the ground in continental climates, but might lead to a cooling of the ground in maritime climates. However, the magnitude of warming/cooling as well as delays in freeze-up depend on the intensity of the rainfall events, calling for accurate representation of these events in larger-scale models.

Further, the timing of heavy rain events may significantly affect the soil temperature response. While our model results suggest that early summer rain mostly leads to a warming effect in all four scenarios, later rain events may mostly have a cooling effect on the ground, even in deeper layers (see Fig. 3a). Longer-term experiments and multi-year observations and experiments would increase our understanding of how severely a single year with heavy rainfall events in summer will change the ground thermal regime not only within the same year, but also in subsequent years and how repeated wet summers can influence the system response. Few studies presently address this phenomenon on an experimental scale[17,38,44]. We conducted a brief multi-year model analysis, assuming a single year and four consecutive years with heavy summer rainfall events followed by 10 years of baseline rainfall conditions. After a single heavy rainfall summer, we find lag effects and even an inversion of the warming effect in the years after a heavy rainfall summer (see Supplementary Text 3.3). Depending on the climate case, the rebound time to <0.1 °C temperature difference between the HR and the reference case in the subsoil can be between two (cold-wet) to eight (warm-wet) years of average conditions after a heavy rainfall summer (Fig. S14a). If several heavy rainfall summers occur in sequence, a dry climate might change towards a wet climate system response (Fig. S14b, cold-dry), and wet climate systems might experience an enhanced cooling as response to several heavy summer rainfall years. On the other hand, the rebound (<0.1 °C temperature difference) in wet climates is significantly faster (3 years in cold-wet and 7 years in warm-wet) than in the dry climate cases, where the temperature difference after ten years is still larger than 1.5 °C (Fig. S14b, cold-dry and warm-dry).

Besides climatic factors, the landscape can influence the response of the active layer thermal regime by changing the path of water flow. Topography, hydraulic conductivity, and geomorphic features such as polygonal tundra can cause differences in moisture distribution and hence change how ground temperatures react to heavy summer rainfall for example through surface and subsurface runoff in sloped terrain, ponding of water in e.g., low-centered polygons, and runoff through trough networks in polygonal tundra landscapes[8,27,55]. Our main results have shown that differences in soil moisture affect the response to heavy rainfall events. Soil moisture further depends on soil properties, water retention capacity, permafrost ice content, thermokarst development, vegetation, snow depth, etc., which thus may have an impact on how heavy rainfall will affect the ground thermal regime. How fast water can infiltrate, vegetation interception, how much water the soil can hold, and how it affects heat conduction and advection can have highly non-linear effects on the overall response to heavy summer rainfall[10,14,56].

To address the sensitivity of our model results to differences in soil composition, and without changing any of the physical soil properties individually, we used the model to investigate the importance of the organic layer thickness to address some of the potential important

influences mentioned above. The organic layer has a significant influence on water redistribution and thermal insulation of the soil and can hence alter the effect of heavy summer rainfall. The sensitivity results show a modest influence of organic layer thickness on the overall temperature effect (see Supplementary Text 3.4 and Fig. S15) but indicate that mineral soils without an organic layer may be affected less than soils with an organic-rich top layer. Further, variability in vegetation may influence the rates of evapotranspiration, which affects the amount of water ultimately reaching the subsurface. For example, tundra shrub vegetation has been found to intercept 15–30% of incoming rainfall[57]. While our model accounts for bare-ground evaporation, transpiration is not explicitly accounted for in its current configuration.

In order to start addressing the system behavior holistically, small scale field experiments will largely help disentangle the effects of overall climate change. Together with a calibrated model, it is then possible to make more accurate assumptions about a site specific system response, which will greatly improve our understanding of multiple facets of climate change on large-scale permafrost thaw.

To further validate our hypothesis that permafrost environments in continental climate are particularly vulnerable to future increased precipitation through changes in the soil thermohydraulic properties, longer term field monitoring will be necessary. Precipitation manipulation experiments in permafrost landscapes have previously been conducted successfully but different effects were observed based on the experimental setup and duration[17,38,44]. This calls for standardized observation protocols to quantify effects of heavy rainfall events or irrigation treatments on permafrost soil thermal regimes across sites. A general guide on how to conduct such an experiment can be found in ref. 58. In addition to the exploration of governing physical mechanisms that determine the effect of summer precipitation on the ground temperatures through modeling, our study provides useful information on where and how to measure the soil temperature response in field experiments. Combining standardized irrigation experiments with local- to large-scale model simulations will help to holistically address the effect of future heavy summer rainfall on permafrost degradation under different climatic conditions as well as different landscapes and soil types.

The question of whether future heavy rainfall in summer will warm or cool the ground in permafrost landscapes is ambiguous and depends on a multitude of interactions. With our study, we have furthered the understanding of the impacts of climate change on permafrost landscapes and identified potential gaps in the design of field experiments. Going forward, more standardized protocols and more field-based evidence across different climatic conditions such as continental and maritime regions are required to increase confidence in the various reasons for the diverging effects observed in the literature and their impacts on large-scale permafrost thaw. Despite the wide range of environmental controls on the soil thermal regime, we found compelling evidence that the response of ground temperatures in permafrost regions to heavy summer rainfall depends on climatology. Sites with continental climates were found to be more vulnerable to accelerated permafrost thaw with increased summer precipitation. On the other hand, permafrost soils in more maritime climates showed cooling following increased summer precipitation, which may potentially attenuate the effect of increasing air temperatures to some extent. A significant amount of ice- and carbon rich permafrost is located in continental permafrost regions, posing an increased risk for carbon release and hence for an increase in the permafrost carbon feedback and global climate change.

## Methods
To address the question weather heavy summer rainfall has a warming or cooling effect on ground temperatures in permafrost landscapes, and whether this may be related to climatological contrasts, we

combined several methodological approaches. We first reviewed relevant literature on the topic to collect information about observed effects. Additionally, we performed statistical analysis on relations between observed rainfall responses (warming or cooling) and climatological setting of the study sites found in the literature. Lastly, we performed model simulations with a state-of-the-art numerical model to further investigate the driving physical mechanisms leading to ground warming and cooling using generic climate scenarios based on the results of the reviewed literature and the statistical analysis as well as publicly available weather data.

## Literature review

To obtain an overview over the current state of research, we conduct a scoping review[59] of studies reporting effects of heavy summer rainfall on the permafrost soil thermal regime. We initially searched for relevant literature by restricting our search to the impact of summer/liquid precipitation on the active layer in permafrost affected landscapes using a database search with key words (See Supplementary Text 1.1.1 for key string and databases), and extended our search strategy using snowball searching and Google Scholar alerts. We screened literature and selected eligible studies using PICOS Criteria[60]. In summary, we considered studies that report effects of heavy rainfall events on active layer thickness, thaw depth or soil temperatures in permafrost soils, using either field experiments, monitoring under natural rainfall variability or model studies (see Supplementary Text 1.1.2 for full PICOS Criteria). We additionally considered articles in which the temperature effect of rainfall was not directly investigated but observed as a by-product of a different research question (e.g., biogeochemistry focused). In several cases, we retained studies that report the effect of only active layer soil moisture on ground temperature differences if the work provides complementary insight into our main research question. Our literature search strategy is presented as a PRISMA diagram in Fig. S16. In total, we found 22 eligible articles that address the topic of heavy summer rainfall effects. Table S1 lists all articles considered in this work categorized by the main observed effect of summer rainfall on active layer temperatures.

We used a vote-counting procedure[61] to subdivide the identified studies into response classes, where we distinguished studies that report warming effects, cooling effects, divergent (both) effects or no effect of heavy rainfall, on either active layer thickness, thaw depths or soil temperatures. We assigned reported effects to response classes based on a rule set defined in Supplementary Text 1.2.1, where increases or decreases in active layer thickness, thaw depth or ground temperatures are considered "warming" and "cooling" effects, respectively. In cases where authors reported divergent effects (both warming and cooling) based on depth of soil temperature observation (e.g., cooling of topsoils but warming of subsoils[17,40,41]), the effect reported at the deepest available soil depth was used, as this was assumed to be most representative of processes at the thawing front. Sites where both effects were observed based on other covariates, for instance, based on heterogeneity among subsites[30,45], were reported as "both" effects. Only one study reports no effect[28].

## Statistical analysis and continentality index

We hypothesized that observed heterogeneity in the response of active layer temperatures to heavy rainfall events in current literature may be related to climatological contrasts. We, therefore, tested whether observed responses reported in various studies, resulting from the vote-counting procedure, were related to climatological contrasts among sites. To reduce variability in reported effects due to differences in study design and time span of observation, we implemented a second round of screening of the identified studies. For statistical analysis of association between reported effects of the selected studies and local climatic data, we narrowed the data selection to those studies that report field-measured data of soil thermal

dynamics (soil temperatures, thaw depth or active layer thickness), within the same year as experimental treatment or naturally occurring heavy rainfall, and at spatially explicit locations. For consistency, we disregard studies reporting effects of only soil moisture increases[30,41], only next-season effects[39] or only modeling results[16,35,36,42,45] (see additional PICOS criteria for statistical evaluation in Supplementary Text 1.2). We acknowledge that this approach does not resolve all methodological differences among studies, and that differences in rainfall event or treatment magnitudes, differences among experimental and monitoring studies and use of different metrics (e.g., thaw depth and soil temperatures at variable depth) may influence our results. Still, we expect that the identified patterns provide a preliminary insight into potential contrasts in rainfall response of permafrost across climatological contrasts.

For the remaining 14 sites, we extracted monthly temperatures and precipitation data from ERA5 reanalysis data for the period 1991–2020[62] to calculate average temperature and total precipitation in winter (December–February), spring (March–May), summer (June–August) and fall (September–November).

Apart from seasonally averaged temperature and precipitation data, we calculated Conrad's Continentality Index (CCI) following Stonevicius et al.[63] for each site over the period 1991–2020. Conrad's continentality index characterizes a place as "oceanic" to "continental" based on the maximum and minimum annual temperature and the latitude at which the place is located (Eq. (1))

$$CCI = \frac{1.7\,(T_{max} - T_{min})}{\sin(\phi + 10)} - 14 \tag{1}$$

where CCI is the Conrad's continentality index, $T_{max}$ and $T_{min}$ are the average annual maximum and minimum air temperatures, respectively, and $\phi$ is the latitude. Minimum and maximum air temperatures are based on the long-term (30 years) average.

We then assessed whether different rainfall responses were associated with contrasts in site climatology using a combination of exploratory (PCA) and inferential (Wilcoxon rank sum test) statistics. First, we ran a PCA on site climatological data (averaged temperatures and seasonal total precipitation per season and CCI) and visually assessed patterns of reported effects of heavy rainfall on the soil thermal regime against the first two principal components using a biplot. Secondly, based on the patterns evident from the biplot (Fig. 2a), we evaluated whether CCI and winter precipitation differed among sites with different response classes (warming, cooling, both or no effect) using box- and jitter plots. Due to the low number of sites that showed "no effect" ($n = 1$), we only tested for significance of difference in continentality index and winter temperature between sites that showed warming ($n = 7$) and sites that showed cooling ($n = 6$) using a Wilcoxon rank sum test. We chose non-parametric tests to account for relatively small sample sizes. Lastly, we checked whether spatial clustering of the sites resulted in spatially correlated residuals using Moran's I. We set our significance criterium at $\alpha = 0.05$.

## Model simulations

To further test the identified patterns of warming and cooling responses across climatological contrasts, we use a permafrost thermal-hydrology model. The simulations are performed with the Advanced Terrestrial Simulator (ATS v1.2, Coon et al.[46]), a physics-based, fully coupled surface-subsurface energy and mass transfer numerical model. In its permafrost hydrology configuration[64] it has previously been successfully applied in simulating a variety of permafrost systems in the Arctic[17,54,65–68]. Due to its physics-based nature, ATS is a powerful tool to explore small scale processes and their effects on energy and mass fluxes. ATS accounts for both variably saturated as well as variably frozen ground when simulating the permafrost thermal regime. It also accounts for a variety of important physical processes

including snow processes such as densification and aging, hydrological processes occurring in 1D, 2D, and 3D spatial model setups, and soil thermal processes including heat advection in both the lateral and vertical direction as well as cryosuction. For further details please refer to ref. 46 and examples of previous applications of the model as mentioned above.

In our model study, we used a 1D column model with a depth of 40 m. We chose a depth of 40 m to ensure that the bottom of the domain is below the depth of zero annual amplitude and the bottom boundary condition does not influence the thermal dynamics of the active layer. The top 0.2 m are initially defined as an organic layer but are later changed for sensitivity analysis to 0.5 m and 0 m depth, while everything below represents a mineral soil (for soil physical properties, please see Table S4). The column mesh is discretized into several vertical cells, which are small near the surface of the domain (2 cm thickness within the active layer) and increase in thickness with increasing soil depth (max. ~ 2 m thickness at the bottom). This way we ensured to adequately resolve the relevant processes in the active layer. The sides of the domain are defined as no-flow boundaries. On the bottom of the domain, a no-flow, constant temperature boundary condition of −9 °C is applied. The surface boundary condition is described by the surface energy balance (SEB). Here, the SEB is composed of daily values for air temperature, relative humidity, wind speed, incoming shortwave radiation, and liquid and solid precipitation (Fig. S1) and is the only boundary condition that varies throughout our different climate scenario model setups.

To obtain a physically consistent thermo-hydrodynamic state to initialize the model, a sequence of steps is required before running transient simulations. First, in a 1D column, we define a water table at a target depth. Second, the entire column gets frozen from the bottom and due to the volume expansion of water freezing into ice, the initial water table needs to be set at a depth that causes the permafrost table to be at the surface after freezing. After the second step, the frozen column serves as initial condition for the spinup (third initialization step), in which the daily weather forcing is applied to the surface, repeating the same year of weather conditions for 100 years to achieve an annually periodic steady state. This way, we ensure the system is physically consistent and can be used as initial condition for the transient simulation runs (fourth step). For each of the four climate scenarios, we run the model first for one year with average conditions and then change for one year to the heavy rainfall scenarios for each of the climates. For result analysis, we use the model output produced in the year of heavy rainfall in each scenario (same-year effects). To present temperature responses to heavy rainfall, we distinguish between topsoil and subsoil. The topsoil is defined as 25% of the maximum annual thaw depth (active layer thickness), while the subsoil is defined as 75% of the maximum annual thaw depth. In the post-processing of model results, topsoil and subsoil depths are defined as the closest mesh element to the depth 25% and 75% of the maximum annual thaw depth. We chose 25% and 75% as representative for top- and subsoil, respectively, as compared to a static depth in each climate scenario, as the ALT varies between different climates. Defining the top- and subsoil in this way allows for enough distance to the surface as well as the permafrost table to avoid boundary effects from the surface energy balance and the presence of the permafrost table, and focus on the effects of the active layer thermal regime. Maximum annual thaw depth is different between all scenarios and hence topsoil and subsoil absolute depths vary slightly and are noted in Table S5. Temperature differences at additional depth layers (surface to ALT, in 20% steps) are shown in Fig. S17.

### Climate scenarios

Our climate scenarios (continental to maritime climate) are generic time series with daily values for one annual cycle that is repeated over the simulation time. To be able to represent these different climates, we retrieved weather data from publicly available weather station observations in Arctic, Sub-Arctic, and alpine permafrost landscapes. Due to the input variable requirements in ATS, which are needed to adequately represent hydro-thermal dynamics, we had to limit our selection of weather observations to observations providing all or most of the variables over at least several years. Individual time series providing all variables except solar radiation were used and complemented with ERA5 solar radiation data[62]. Table 1 lists information about the stations from which data was retrieved. Depending on the length of observations, we calculated daily averages over the time span of seven to 70 years for each day of the year (DOY). For each of the sites, we then calculated the average annual air temperature, mean annual summer temperature and precipitation from June to August, mean annual winter temperature and precipitation from December to February, and yearly cumulative snow and rain precipitation. Based on the observed long-term average temperatures and the annual air temperature amplitudes between the coldest and the warmest day, we determined representative climates for a warm-dry, a cold-dry, a warm-wet, and a cold-wet summer climate. A warm climate represents

**Table 1 | Information about the sites that observational weather data has been retrieved from in order to create the synthetic climate scenarios including information about location, observation length, data source, long-term average temperature ($T_{avg}$) and annual temperature amplitude ($T_{amp}$), long-term average summer precipitation sums (summer P), average relative humidity (RH), and average wind speed (wind)**

| Place name | Lat Lon | Observation length | ERA5 | Ref | $T_{avg}$ [°C] | $T_{amp}$ [°C] | summer P [mm] | Avg. RH [%] | Avg. wind [m s⁻¹] |
|---|---|---|---|---|---|---|---|---|---|
| Adventdalen | 78.19N 15.81E | 2013–2020 | No | 69,70 | −3 | 26.4 | 96 | 73.0 | 5.1 |
| Alert | 82.5N −62.35W | 2004–2019 | No | 71,72 | −15.5 | 39.4 | 54 | 77.3 | 2.9 |
| Barrow | 71.2N −156.5W | 2000–2020 | No | 73,74 | −8.9 | 34.8 | 74 | 67.0 | 5.8 |
| Disko Island | 69.68N −53.73W | 1993–2020 | No | 75 | −2.5 | 25.1 | 196 | 73.2 | 3.4 |
| Imnavait | 68.76N −149.41W | 1981–2018 | No | 71,76 | −7.2 | 47.9 | 224 | 74.3 | 2.9 |
| Samoylov Island | 72.38N 126.48E | 2005–2019 | No | 77 | −11.7 | 45.9 | 93.7 | 80.9 | 4.3 |
| Kytalyk | 70.83N 147.5E | 2000–2020 | Yes, $Sw_{in}$ | 78 | −13.5 | 46.9 | 96 | 80.9 | 4.0 |
| Tibet | 34.23N 82.44E | 2007–2019 | No | 79 | −2.5 | 27.3 | 323 | 52.2 | 3.7 |
| Yakutsk | 62.01N 129.66E | 1966–2020 | Yes, $Sw_{in}$ | 78 | −8.7 | 61.5 | 161 | 67.9 | 1.7 |
| Marre Sale | 69.71N 66.67E | 1950–2020 | Yes, $Sw_{in}$ | 78 | −7.4 | 33.3 | 140 | 87.0 | 6.2 |
| Zackenberg | 74.47N −20.53W | 1995–2020 | No | 75 | −8.8 | 30 | 58 | 71.3 | 2.8 |

Note that these locations do not align with the study sites in the literature review displayed in Fig. 1. If ERA5 data was used to complement the observational dataset, it is indicated with "yes" followed by the variable added.

**Table 2 | Summary of the synthetic climate scenarios**

| Climate scenario | $T_{avg}$ [°C] | $T_{amp}$ [°C] | C [days] | rain sum [mm] | snow sum [mm] | rain [mm d$^{-1}$] | snow [mm d$^{-1}$] |
|---|---|---|---|---|---|---|---|
| Warm-dry | −11.7 °C | 45.9 °C | 260 | 60 mm | 50 mm | 0.50 | 0.20 |
| Cold-dry | −8.8 °C | 30 °C | 260 | 60 mm | 50 mm | 0.54 | 0.20 |
| Warm-wet | −11.7 °C | 45.9 °C | 260 | 300 mm | 50 mm | 2.50 | 0.20 |
| Cold-wet | −8.8 °C | 30 °C | 260 | 300 mm | 50 mm | 2.72 | 0.20 |

Average annual temperature ($T_{avg}$), average annual temperature amplitude ($T_{amp}$), and the phase shift factor (C, see Eq. (2)) are needed for the sine curve that describes temperature, rain- and snow sum represent the sum of summer and winter precipitation, respectively, rain (mm d$^{-1}$) and snow (mm d$^{-1}$) indicate the daily rain- and snowfall rate, respectively. Daily precipitation amounts are different for all scenarios due to the difference in infiltration period (rain: air temperature > 0 °C, and snow: air temperature ≤ 0 °C) caused by differences in air temperature in the warm and dry scenario. Both temperature and precipitation values shown here represent the upper and lower range of observed climatic conditions at weather stations listed in Table 1.

a warm summer, but relatively cold winter (continental climate, e.g., Samoylov Island, see Table 1), while a cold climate is represented by a relatively cold summer, but mild winter (maritime climate, e.g., Zackenberg, see Table 1). To derive daily air temperatures T(doy), we used a slightly offset symmetrical temperature sine curve describing the warm and cold climate based on average temperature and annual amplitude found in Samoylov Island and Zackenberg as representative baseline temperature conditions for a continental and maritime climate, respectively, as follows

$$T(\text{doy}) = T_{\text{avg}} + \frac{T_{\text{amp}}}{2}\sin\left(\frac{2\pi^*(\text{DOY}+C)}{365}\right) \qquad (2)$$

where $T_{\text{avg}}$ is the average annual air temperature, $T_{\text{amp}}$ is the air temperature amplitude, DOY is the day of the year and C represents the phase shift factor (here: 260).

Precipitation is represented by sums over the infiltration period for each, rain and snow (rain: air temperature >0 °C, and snow: air temperature ≤0 °C) and then distributed to equal amounts for each day in the same period. Due to differences in air temperature development between the cold and the warm scenarios, this leads daily precipitation values not only being different between the dry and wet scenario, but also between the cold and warm scenarios. For the model forcing snow precipitation was set to 50 mm per infiltration period. We chose a value that represents comparably little snowfall to avoid additional complexity due to snow melt and to better extract the effect of rainfall. Total summer rainfall is set to 60 mm and 300 mm for the dry and wet scenario, respectively, and covers the lower and upper range of observed average summer precipitation observed at the weather stations in Table 1. Table 2 lists the key values used to describe the sine curve as well as precipitation sums in mm and daily precipitation rate in mm d$^{-1}$.

The remaining variables needed to inform ATS were set as follows: Daily incoming shortwave radiation $G_{\text{doy}}$ is calculated using a similar sine curve as used to calculate daily temperature:

$$G(\text{doy}) = G_{\text{avg}} + \frac{G_{\text{amp}}}{2}\sin\left(\frac{2\pi^*(\text{DOY}+C)}{365}\right) \qquad (3)$$

with G(doy) being the daily incoming shortwave radiation, $G_{\text{avg}}$ being the average solar radiation (here: 116 W m$^{-2}$), and $G_{\text{amp}}$ representing the annual amplitude in solar radiation (here: 221 W m$^{-2}$), resulting in an annual minimum of 6 and a maximum of 227 W m$^{-2}$, which covers a range of Arctic winter insulation to sub-Arctic summer insulation. Relative humidity was set to a constant value of 80% and wind speed was set to 4 m s$^{-1}$. These values are within the range of representative values commonly observed in permafrost landscape climates (such as at the weather stations listed in Table 1) and, by keeping them constant, help to isolate the effect of only temperature and precipitation on the ground thermal regime. Together, this dataset comprises the surface energy balance forcing used for the model simulations described in the above section.

To simulate heavy rainfall, we use recurrence intervals of extreme precipitation from the weather station data described above. We use a 100 year recurrence interval calculated using the Weibull distribution and extrapolating the recurrence intervals to a 100 year period using a linear regression model. The observed recurrence intervals are based on the time period available from the weather data and describe in which interval single-day extreme events are occurring. To simulate a present-day heavy rainfall event with a 100 year recurrence interval, we used the last value of the linear regression model for each of the weather observation sites and for each month. We then use the average over all observation sites to define a single average 100-year recurrence heavy rainfall event in each, June, July, and August. The resulting values are 32 mm in June, 45 mm in July and 50 mm in August are applied to all climate scenarios disregarding their base precipitation rate. In the ATS forcing dataset we then replace the precipitation values on 15th of each, June, July, and August with the respective heavy rain event values. With a total of 127 mm added summer rainfall, this represents a three times wetter summer for the dry climate scenarios and a roughly 36% increase in summer precipitation for the wet climate scenarios. We choose to add a static value to all climate scenarios to account for a heavy rain event independent of base precipitation rates. Despite active layer development often reaching into September in various regions[17], we did not add a September event due to potentially confounding interactions between rain- and snowfall due to low air temperatures during late summer/autumn. In order to account for the model sensitivity towards varying intensities in rainfall magnitude and temporal distribution, we also created scenarios that address equally/uniformly distributed increased summer rainfall (see Supplementary Text 3.1) and varying magnitudes of heavy summer rainfall events (see Supplementary Text 3.2). In the uniform scenario, summer rainfall is generally increased by 50% and therefore covers the other end-member along a heavy-rain-event to general-precipitation-increase gradient. In the scenarios with varying magnitude in rainfall events, we simulated 50% stronger heavy rainfall events as well as 50% less intense heavy rainfall events.

## Data availability
The model input and forcing files generated in this study have been deposited in a Zenodo repository at https://zenodo.org/record/7957042. Source data are provided with this paper.

## Code availability
The Advanced Terrestrial Simulator (ATS) (https://doi.org/10.11578/DC.20190911.1; Coon et al.[46]) is open source under the BSD 3-clause license and is publicly available at https://github.com/amanzi/ats (last access: October 2022). Simulations were conducted using version 1.2.

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

## Acknowledgements

Financial support for A.H. and A.F. for this research has been supported by the Svenska Forskningsrådet Formas (grant no. 2017-00736) and by the Bolin Center for Climate Research. R.Í.M. acknowledges funding

from the Netherlands Polar Program of the Dutch Research Council under grant number "ALWPP.2016.008". Computations were enabled by resources provided by the National Academic Infrastructure for Supercomputing in Sweden (NAISS), partially funded by the Swedish Research Council through grant agreement no. 2022-06725.

## Author contributions

A.H. and A.F. conceptualized the study, A.H. and R.Í.M. developed and designed the methodological approach and reviewed relevant literature. A.H. designed and conducted model simulations and analysis with support from A.J.K. and A.F. R.Í.M performed statistical analysis. A.H. wrote the manuscript with input from R.Í.M, A.F., and A.J.K.

## Funding

## Competing interests

The authors declare no competing interests.
