## [Peer Review File · Nature Communications]

Continentality determines warming or cooling impact of heavy rainfall events on permafrostREVIEWER COMMENTS

Reviewer #1 (Remarks to the Author):

Summary

In their study, Hamm et al. investigate the effect of increased summer precipitation on the soil thermal regime and permafrost thaw in permafrost regions of the Northern Hemisphere. Therefore, the authors combine a literature review with a physics-based numerical model experiment to identify and evaluate the mechanisms, as well as important climatic factors affecting the ground thermal response to heavy summer rainfall in Arctic and Alpine permafrost environments.

Their study suggests that heavy summer rainfall leads to ground warming and likely increased permafrost thaw in warm, dry, and continental climate permafrost regions, whereas it leads to a slight cooling and decreased permafrost thaw in cold, wet, and maritime regions. Hence, Hamm et al. highlight that warm, dry and continental climate regions are especially vulnerable to rapid permafrost degradation, and as these regions tend to store relatively high amounts of soil organic carbon, increased summer rainfall potentially accelerates the permafrost carbon feedback.

General impression

Hamm et al. present an interesting and timely investigation of an important question of high relevance to a broad spectrum of the scientific community. The combination of the literature review with a numerical modelling experiment is innovative and overall, well done.

Specifically, whereas the literature review is based on a relatively small and variable set of studies hampering strong and robust inferences, the combination with the numerical model experiments adds substantial depth and rigor and enables the authors to derive useful insights & predictions in a case where not a lot of in-situ data is yet available. I think the identified mechanisms and climatic factors affecting summer rainfall - ground temperature - permafrost thaw relationships lay an important basis for future studies on this important issue. The study is generally well written, for the most part it is relatively clear and concise, and the figures are informative and well designed.

Hence, in principle, I recommend the manuscript for publication. However, there are a set of shortcomings, predominantly related to the clarity and robustness of the study that I suggest need to be addressed beforehand.

In particular, I find the literature review methods and results are not described with sufficient clarity and detail to understand the collection and analysis process, and therefore, I'm not yet entirely convinced by the results, even though I find the general approach and statistical tests appropriate. Also, it would improve the clarity of the manuscript a lot, if the authors more clearly highlighted the focus of the study ("increased" vs. "extreme" rainfall), and more clearly described the definition of important terms (such as "topsoil" and "subsoil") and the reasoning behind definitions and parameter settings in the modeling part. More detailed recommendations are described below.

Detailed comments

Focus of the study & definition of rainfall scenarios

- Even though the title nicely captures the focus on "heavy rainfall", in the Introduction section, it is not very clear if this study is going to be about increased summer rainfall in general or about extreme precipitation events, i.e. "heavy rainfall". I think it would improve the clarity of the study if the authors could highlight the focus on "heavy" or "extreme" rainfall events even more throughout the manuscript and provide a clear definition of what they mean by "heavy rainfall" in the context of this study.
- Maybe you could add a column in Table S4 describing the amount of increase in rainfall that was investigated in the different studies of the literature review dataset?
- Finally, it would help to understand the study if you more clearly introduced the main precipitation scenarios of interest (i.e. "HR" and "ref.case" scenario) as well as the third precipitation scenario ("uniform precipitation increase") that you investigated in your study.
- Some examples that caused confusion for me:

- o on lines 37-38 it is described that studies assessed effects of “heavy summer rainfall”, whereas on line 39 it is about general “summer rainfall”?
- o Similarly, on lines 244 ff. the authors describe that the studies investigated soil responses to “heavy rainfall events”, even though in your literature search you just searched for “rainfall” and “summer precipitation” – did you only select studies that were concerned with “heavy rainfall”? Or did they all do so by chance?
- o on lines 116-118: Could you explain here or in the Methods by how much you increased the heavy rainfall and why you chose that amount and not another one? I.e. could you explain your reference scenario and “HR” scenario more clearly?
- o on line 210 the authors mention the “uniform” increase of precipitation scenario a bit out of the blue, without introducing the reason of this scenario very well. Since the conclusions are important (soil response does not depend on rain frequency) I think it would be worth to better introduce this scenario with the other two (“HR” and “reference”).

Definitions of topsoil and subsoil

- The division of subsoil and topsoil is quite important in this study. I think it would be important to explain and define these concepts upon first mention in the Introduction section already (line 52).
- As I understand it, the authors define “topsoil” as the soil column exactly at the depth of 25% length of the maximum active layer thickness length (ALT) and “subsoil” as the soil exactly at depth of 75% of maximum ALT? Please describe this more clearly and it would be helpful if the authors added
 - 1) an explanation of why these thresholds were chosen and why they are adequate, if possible, citing previous research or previous analyses of the literature data at hand
 - 2) a discussion of the repercussions of that choice – i.e., does the choice of this relative definition versus fixed absolute boundaries for topsoil and subsoil affect how the results need to be interpreted?
 - 3) the magnitude of the variance of topsoil and subsoil absolute depths among study sites and modelled climate scenarios (cf. line 311)? Do they vary a lot? Could you provide an argument or evidence that this does not matter for your conclusions?

Literature review description

- The literature review is not very well described, neither in the Results nor in the Methods section.
- In the Methods section, it would increase the clarity of the study, if you described
 - 1) Where (on which databases) you searched for the literature
 - 2) Provide the exact key-word string and date of the search
 - 3) Provide the criteria for inclusion or exclusion of literature
 - 4) Ideally, it would be possible to describe the initial number of studies and studies excluded at several stages of the screening and reading process (cf. Liberati et al. 2009)
 - 5) Did studies you collected report on soil temperature effects to heavy rainfall events (i.e. as you define it as extreme rainfall?) or just rainfall in general? Did you consider warming measured in the same year or at the end of the season or even warming in the next year?

Liberati, A. et al. The PRISMA statement for reporting systematic reviews and metaanalyses of studies that evaluate health care interventions: explanation and elaboration. *J. Clin. Epidemiol.* 62, e1-34 (2009).

- In the Results section, literature review findings tend to be described in a qualitative way, and it would add clarity and rigour to your study, if you could be a bit more concrete and quantitative in your description and add relevant references to literature or Figures/Tables in your study. For example:
 - o I think it would add to the clarity of the text if you could provide references for claims on lines 31-33
 - o Also, on lines 37-38, you describe that “in the literature, opposing effects of heavy summer rainfall on ground temperatures have been reported in different studies...”, however there are no references. Here it would be good to add such references to the corresponding studies.

- o Lines 49-51: “Effects of heavy rainfall on permafrost has seen increasing attention in scientific literature in recent years, with an increasing number of studies that add a modeling analysis to their methodology. A detailed overview of the identified studies is presented in Table S2.” – Here it was not clear to me what it means to “add a modeling analysis”?
- o Lines 57-60: Here some clarity needs to be added: how did you define topsoil and subsoil? Did you take your own definition or according to the definition of each corresponding study? Also, it would be adequate to give a reference to the claim “As subsoil temperatures are decisive for thaw depth development”.
- o For example, when you describe the studies on line 57 ff., it would be nice to reference the corresponding studies which are alluded to. For example, I was not sure if the “two studies that address effect of wetter soil conditions through soil moisture,...” are now in addition to or if they are part of the 20 studies found in the literature review? Which ones are they in Table S2?
- o Lines 60-62: could you please add the references of the studies you allude to here?
- o Similarly, on line 69, you describe “In general, studies that report thermal effects over multiple soil depths find warming of subsoils and cooling of the topsoil” – It would be important to add a reference to one of your tables or analyses so that this claim can be backed by a bit more rigor.
- o The same on line 70: “Not all the studies report temperature response but rather report changes in ALT..” which studies are alluded to here? Please be a bit more precise.
- o Lines 80-81: “We observed...higher continentality...in sites that showed warming effects of heavy rainfall events” – Did all the studies report effects of heavy rainfall or just increased rainfall? By how much did rainfall increase in these cases?
- o In Table S2: It seems that at least some observational studies are experiments – could you add this nuance?

Literature review analysis

- Several aspects of the literature review analysis, especially the data aggregations appear a bit arbitrary, are vaguely described and not well justified. If you could add some justifications and explanations of the decisions you made, that would make the analysis more convincing.
- o On line 72-73 you describe that rainfall effects on soil temperatures can strongly depend on the duration and depth of monitoring of the soil temperatures. Have you considered this when statistically comparing the results of the literature review studies? If not, please explain why.
- o The literature review covers a relatively small set of studies with different methods and time intervals of investigation. It would add to the clarity of the manuscript if information about measuring depth, magnitude of precipitation increases and timescales of the literature studies would be indicated in Table S4, for example.
- o On line 76, It was not easy for me to understand why you analyzed climatological data only from 16 of the 20 studies? Please clarify.
- o I think the type of analysis conducted with the literature data is more similar to a vote-counting approach rather than a meta-analysis, which would require effect sizes extracted from the collected studies (cf. Koricheva et al. 2014 and Gurevitch et al. 2018), hence, consider rewriting.
- o Lines 249-251: I think it is not adequate to simply exclude literature review studies that show no clear cooling or warming effect. Please provide a convincing reason to exclude this data from your analysis, if possible with reference.
- o Similarly, on lines 263-264 you describe that results from sites less than 100 km apart were “averaged”, even though averaging in this context means (as I understand) ascribing the most frequent value (i.e. in the case of three studies where two showed a cooling [Zhang and Zhou 2021] and one showed a warming [Li 2019], you ascribed a “cooling” effect [lines 266-267]). I have two questions: 100 km seems a large distance to me that not necessarily causes a spatial pseudo replication: on what basis did you decide to average sites less than 100 km apart? Could you add a reference or chain of arguments that supports this distance threshold? Additionally, I think it is not adequate to “average” three studies to a cooling effect, when in fact one of three studies did show a warming effect? As in the case above (Lines 249 ff.), it would be more adequate to keep the original studies and their original results in the literature analysis.
- o Lines 252-254: Why did you take the monthly temperatures for the time 1959-2021 to calculate

mean seasonal and annual temperature and precipitation and not a 30-year window closer to the temporal extent of the studies in the literature, e.g. 1990-2022? Additionally, on lines 255-256: Why did you choose again a different time (of 1981-2010) for the continentality index?

Koricheva, J., & Gurevitch, J. (2014). Uses and misuses of meta-analysis in plant ecology. *Journal of Ecology*, 102(4), 828-844.

Gurevitch, J., Koricheva, J., Nakagawa, S., & Stewart, G. (2018). Meta-analysis and the science of research synthesis. *Nature*, 555(7695), 175-182.

Non-climatic factors

- As you also describe in your study, not only climatic factors but also local characteristics of the landscape, vegetation, permafrost, and soil properties can affect the ground temperature response to increased rainfall. Is there a reason why you did not include such local soil type (e.g. water retention capabilities, organic layer thickness), permafrost type (e.g. ice-content) or vegetation type into the PCA of the literature review analysis (e.g. described on lines 74-79) and subsequent modelling analysis? The results would be more convincing if you could provide an argument why you did not (need) to consider soil type, permafrost ice-content or vegetation types for the main part of your analysis.
- Also, could you add a description of how you identified the factors important to consider in your study: i.e. did you identify the climatic predictors from the collected literature? And which variables exactly did you test? For example, in line 84 you describe: "No significant contrasts in other climate variables were observed..." Could you provide a description of these other tested variables and the method how these were identified?
- I think it is important to discuss additional, non-climatic factors affecting the soil thermal response to increased precipitation, such as on line 207 ff.

Temporal extent of interest

- I think it would greatly improve the clarity of the manuscript if the temporal extent (a season, a year or several years?) and resolution (daily, monthly, yearly?) of the collected literature and the modelling study was clearly described, if possible, in the Introduction or Methods section. As I understand, the modelling experiment for example is focused on within-season changes of the soil thermal regime due to heavy rainfall events, but no lagged or multiyear effects? Please clarify.

Modeling analysis: selection of scenario and parameter values

- I think the model experimental setup with the four different climate scenarios are well conducted and informative. However, there are some decisions on modelling scenarios and data processing that I find not very clearly explained. Specifically:
 - o Line 210: "We also assessed the impact of heavy rainfall events compared to a uniform increase of precipitation by 50%": this comes a bit out of the blue and it is not clear when and why you did that. It would be nice to better describe that in the Methods and Results sections.
 - o Lines 316 ff.: Why did you choose in-situ weather station data for a small selection of sites instead of calculating a more representative pan-Arctic average & range from ERA5? What were the variables that you retrieved from the stations as input to the ATS? I think it would increase the clarity of the study if you could add these variables to Table 1?
 - o Lines 323 ff.: How did you define what is "representative" of cold and warm climates? Based on quantile or PCA-like analysis of the in-situ station data? Or based on other, literature derived criteria? It would be nice if you could add a more precise explanation here.
 - o Line 330-333: How did you determine "low" and "high" precipitation scenarios? Was this decision based on predictions from previous research? Or based on quantile-analysis from Arctic-weather station data? Cf. also lines 340-342: Please explain why you chose exactly "a 100-year recurrence

interval” precipitation event? Why did you simulate a 100-year recurrence interval event to happen 3 times in a row in June, July and August?

o Table 2: I would replace “MAAT” with “Tavg” and “AMP” with “Tamp” to be consistent with equation 2)

o Lines 335-336: How did you determine the parameters (minimum of six and maximum of 227 Wm⁻²? Are these parameters based on previous analyses or averages for the latitudes of interest? Please clarify. It would also be clearer to describe the exact equation instead of “similar to Eq. 2”

o Lines 337-338: “These values are within range of representative values in permafrost landscape climates and help to isolate the effect of only temperature and precipitation on the ground thermal regime.” – I think that is a valid argumentation – however could you provide a reference to what is representative for permafrost landscapes?

o Line 351-352: The implementation and results of this “relative” increase in precipitation scenario are not well explained, could you be a bit more elaborate on that? Why did you choose a 50% increase in precipitation and not more or less?

o Generally, could you add in your discussion an elaboration on how your model assumptions affect the interpretation of your results? Are “real-world” effects likely to be over- or underestimated? Or do the parameter settings of your model not influence your findings?

Sensitivity analyses

- I was wondering on what basis the precipitation scenarios (cf. Table 2) were chosen and how much the results would change if one would apply a smaller or larger amount of heavy rainfall. Did you explore in that direction? A sensitivity analysis with different magnitudes of heavy rainfall scenarios could shed light on the uncertainty and range of warming/cooling estimates.

- Similarly, it would help to interpret the significance of soil thermal responses to heavy rainfall compared to temperature and continentality. Therefore, one could compare the magnitude of heavy rain-induced differences in topsoil and subsoil temperatures to the general temperature differences among the four climate scenarios, similar as is done for ALT in Figure 3 c).

- Such analyses could help to better interpret the magnitudes and assess the robustness of predicted soil thermal responses to heavy rainfall.

Discussion of mechanisms

- I think that some topics deserve more attention in the Discussion section, in particular:

- o Soil moisture vs. Precipitation: as you already describe on line 61, soil moisture is not necessarily coupled to precipitation. It would be nice if you could elaborate a bit on the precipitation-soil moisture relationship in your Methods and/or Discussion section and explain what assumptions you made about that relationship in your modeling and how that influences the interpretation of your results?

- o E.g. on Lines 147 ff. you describe the dry soils gain a strong increase in thermal conductivity due to increased saturation caused by heavy rainfall – are the modelled conditions realistic and representative for the permafrost areas of the Northern Hemisphere?

- o Timescales: Your analysis mainly focuses on intra-seasonal, “immediate” effects of rainfall. However, as you describe, effects of rainfall on the soil thermal regime can differ according to the timescale of interest (cf. eg. lines 95-98). Hence, it would be nice to add a short discussion on immediate versus cumulative, multiyear effects of increased rainfall on soil thermal regimes and permafrost thaw.

Minor comments

Main text

Lines 71-72: “due to regional or local conditions,” I think this segment is not necessary for the argument at hand and more confusing, therefore I recommend removing it.

Line 114: this claim could be supported by adding a reference to Figure 2d: “...central Siberian permafrost region (Figure 2d)”

Lines 126-127: I don't exactly understand the meaning of this sentence, could you rephrase and clarify?

Lines 236-237: what do you mean by "cross referencing" in this context? How did you do the keyword search and on which platform(s)? What was the exact key-string?

Lines 277: I would add the reference of Coon et al. 2019 here as well.

E.T. Coon, M. Berndt, A. Jan, D. Svyatsky, A.L. Atchley, E. Kikinzon, D.R. Harp, G. Manzini, E. Shelef, K. Lipnikov, R. Garimella, C. Xu, J.D. Moulton, S. Karra, S.L. Painter, E. Jafarov, and S. Molins. 2020. Advanced Terrestrial Simulator. U.S. Department of Energy, USA. Version 1.0. DOI

Lines 290-291: what is the size of the column mesh?

Lines 304-305: exchange "extreme precipitation" with "heavy rainfall" to be consistent with the terminology?

Supplementary Material

Lines 630 ff: could you reference the data you allude to here? Is this from Table 1?

Figure S3: "Differences are displayed as HR case.." – Did you mean "uniform precipitation increase case" instead?

Reviewer #2 (Remarks to the Author):

General comments:

This is a well-written and presented manuscript that will be of interest to the readership. Figures are easy to read and consistent. The results are noteworthy in providing an analysis to the potential role of increasing summer precipitation on permafrost thaw. The data analysis and methodology are well presented and provided in sufficient detail.

Overall the flow of the manuscript is good except I have a comment about the Conclusion section needing a rewrite (below).

I am not sure why the "hook" is the C feedback as there are many other reasons why permafrost thaw is bad for humans, ecosystems, infrastructure, hydrology, biogeochemical processes outside of C, hydrology, runoff, contaminants..... There is too much carbon cycle ramification and not enough of the myriad other reasons to study this. Why the yedoma map? It is not needed.

Fundamentally this paper is about permafrost thaw but there is no mention of "top down thaw" or "near-surface permafrost" which are critical components of the processes. There is also no mention of how/where increased precipitation can initiate or expand lateral thermokarst thaw degradation. Slope failure, lubrication/slumping along thaw fronts facilitated by water, and other processes should be at least mentioned. Some of this could also be in here instead of the yedoma maps and talk of C.

Comments by line #:

11-14: I recommend mentioning carbon briefly but there are also infrastructure, hydrology, agricultural, and ecosystem aspects of increasing top-down thaw of permafrost.

24: of the role

26: local increase where? The Arctic is quite expansive.

90: into two groups

I recommend removing the #1 and #2 and say "into two groups." First,, and then describe that.

The say "The second mechanism is... and explain that. The numbers are confusing.

89-100 Here I wonder if the role of added water in the basal soil leads to the creation of taliks or prevents/reduces the re-freezing of the ground in the following winter.

Figure 3. Maybe put the topsoil as a and the subsoil as b then keep thaw depth c. Makes for more of a "top-down" presentation.

When it rains the temperature in a given area drops (compared for example to warm sunny conditions). Is this addressed in the model? If so, how/where? If not can you provide some sort of

addressal in the text?

192: were observed

206" without an organic layer

210-215: Any sense that heavier rainfall has less residence time due to rapid runoff and thus that rain water does not have as much impact? Is that process accounted in the model?

213: "not very sensitive" is vague.

Conclusion:

At current this is merely a regurgitation of the Discussion main points and the abstract. I recommend moving some of the broader research questions here instead. For example, the roles of the timing of the precipitation, the rate, the organic matter, etc.- to various things that were not addressed. This way you could postulate/hypothesize/recommend these things be studied.

263-268: was the slope, vegetation, or soil type also summarized from the study sites? I realize this may be a lot of work and may not be in all studies but am curious.

The role of slope/rapid runoff is also something to consider at least by providing a sentence on this at the right location (perhaps in the main text).

284: "setpus"

Should this be "setups" or maybe it is a word I do not know.

297: spelling of "thermo"

310: scenarios

321: most active layer measurements are made later than August. This may not be a critical issue with the modeling but is worth noting.

Reviewer #3 (Remarks to the Author):

General Comments:

How enhanced precipitation and global warming can help thaw permafrost is a topic of great interest. Through a meta-analysis of papers from the circumpolar region, the authors assess the current role of precipitation in increasing permafrost temperatures over a wide area and present the impact of the manifested effect in dry and continental climates. This is a new approach. In addition, the numerical model experiments have succeeded in presenting specific processes that show how precipitation can manifest temperature increases in the active layer. These suggestions can be highly appreciated as important basic information when considering various effects of enhanced precipitation due to climate change in the future, such as permafrost degradation, greenhouse gas fluxes, and ecological and hydrological changes. I basically recommend the paper be published, and I really have no fatal comments on the manuscript. It is clearly written and well explained, and the argument is easy to follow. However, I believe two major issues are missing from the content of this paper's discussion based on these two methods.

1) The first point is that the authors evaluate extreme precipitation as a single-year event. The impact on soil moisture and temperature structure in the active layer in a single year would not be significant. The problem is strongly amplified when there is a series of multi-rainfall years. In the central Lena River basin, a series of high summer precipitation years from 2004 to 2008 resulted in the oversaturation of soil moisture in the active layer and widespread deepening and thawing of frozen soil in the active layer. The lack of mention of multi-year effects may be a significant deficiency. I am also concerned about the relationship with the subsequent winter. Late summer rains have been shown to have a cooling effect on deep soils, which may delay winter freezing and consequently contribute to warmer temperatures the following year.

2) Second, although the numerical modeling experiment gave equal amounts of rainfall in July, August, and September, the effect of rainfall on the active layer would change significantly in the summer months due to the budget between precipitation and evapotranspiration. Rainfall in late summer (especially in August and September) may modify the frozen soil's hydrothermal environment. Suppose the assessment is made in the Arctic and subarctic regions. In that case, more attention should be paid to the relationship of the surface water budget between the different vegetation zones (grassland, boreal forest, and tundra).

Specific comments:

Figure 1: It may not affect much, but Sporadic colors are not distinguishable.

Table S2: The study area seems to be listed incorrectly. (E.g., Lena Delta -> central Yakutia in Iijima et al. 2010)

Response letter to the reviewer comments of NCOMMS-22-44895A: Warming or cooling? The impact of heavy summer rainfall on permafrost environments

Comments by the reviewers are written in black font

Our responses are written in blue font

Figures in this response letter are labeled Fig. Rx in addition to the corresponding number in the manuscript, if the Figure is part of the revised manuscript.

Reviewer #1

Summary

In their study, Hamm et al. investigate the effect of increased summer precipitation on the soil thermal regime and permafrost thaw in permafrost regions of the Northern Hemisphere. Therefore, the authors combine a literature review with a physics-based numerical model experiment to identify and evaluate the mechanisms, as well as important climatic factors affecting the ground thermal response to heavy summer rainfall in Arctic and Alpine permafrost environments.

Their study suggests that heavy summer rainfall leads to ground warming and likely increased permafrost thaw in warm, dry, and continental climate permafrost regions, whereas it leads to a slight cooling and decreased permafrost thaw in cold, wet, and maritime regions. Hence, Hamm et al. highlight that warm, dry and continental climate regions are especially vulnerable to rapid permafrost degradation, and as these regions tend to store relatively high amounts of soil organic carbon, increased summer rainfall potentially accelerates the permafrost carbon feedback.

General impression

Hamm et al. present an interesting and timely investigation of an important question of high relevance to a broad spectrum of the scientific community. The combination of the literature review with a numerical modelling experiment is innovative and overall, well done.

Specifically, whereas the literature review is based on a relatively small and variable set of studies hampering strong and robust inferences, the combination with the numerical model experiments adds substantial depth and rigor and enables the authors to derive useful insights & predictions in a case where not a lot of in-situ data is yet available. I think the identified mechanisms and climatic factors affecting summer rainfall - ground temperature - permafrost thaw relationships lay an important basis for future studies on this important issue.

The study is generally well written, for the most part it is relatively clear and concise, and the figures are informative and well designed.

Hence, in principle, I recommend the manuscript for publication. However, there are a set of shortcomings, predominantly related to the clarity and robustness of the study that I suggest need to be addressed beforehand.

In particular, I find the literature review methods and results are not described with sufficient clarity and detail to understand the collection and analysis process, and therefore, I'm not yet entirely convinced by the results, even though I find the general approach and statistical tests appropriate. Also, it would improve the clarity of the manuscript a lot, if the authors more clearly highlighted the focus of the study ("increased" vs. "extreme" rainfall), and more clearly described the definition of important terms (such as "topsoil" and "subsoil") and the reasoning behind definitions and parameter settings in the modeling part. More detailed recommendations are described below.

We would like to thank the reviewer for the extensive feedback to our manuscript and the very constructive comments that have helped to significantly improve the rigor of the study. We have put a lot of work into improving the description of our literature search and the statistical analysis, adding additional sensitivity analysis according to the suggestions, and discussing additional, non-climatic factors that can affect the observed response in our study. For detailed responses, please see our comments below.

Detailed comments

Focus of the study & definition of rainfall scenarios

1. Even though the title nicely captures the focus on “heavy rainfall”, in the Introduction section, it is not very clear if this study is going to be about increased summer rainfall in general or about extreme precipitation events, i.e. “heavy rainfall”. I think it would improve the clarity of the study if the authors could highlight the focus on “heavy” or “extreme” rainfall events even more throughout the manuscript and provide a clear definition of what they mean by “heavy rainfall” in the context of this study.

For the main model analysis, we consider heavy rainfall events. In the literature, we also include studies that are not necessarily looking at individual heavy rainfall events in particular, but overall increased summer precipitation. In the model, we include sensitivity studies (which have further been extended in the revised manuscript) where the rainfall events are spread out more uniformly. We have clarified this in the revised manuscript, and also now consistently use the term “heavy rainfall”.

2. Maybe you could add a column in Table S4 describing the amount of increase in rainfall that was investigated in the different studies of the literature review dataset?

We appreciate the suggestion and have added this information to Table S3 in the revised manuscript.

3. Finally, it would help to understand the study if you more clearly introduced the main precipitation scenarios of interest (i.e. “HR” and “ref.case” scenario) as well as the third precipitation scenario (“uniform precipitation increase”) that you investigated in your study.

We have made attempts to clarify the main simulations and the cases considered for sensitivity studies; please see responses to comments #8 and #36 below.

4. Some examples that caused confusion for me:

5. on lines 37-38 it is described that studies assessed effects of “heavy summer rainfall”, whereas on line 39 it is about general “summer rainfall”?

L37: added “heavy” to “summer rainfall”

6. Similarly, on lines 244 ff. the authors describe that the studies investigated soil responses to “heavy rainfall events”, even though in your literature search you just searched for “rainfall” and “summer precipitation” – did you only select studies that were concerned with “heavy rainfall”? Or did they all do so by chance?

In response to the comments about the literature review description and analysis below, we have improved the documentation about the literature search (see below). A PRISMA diagram (Fig. S2) and PICOS criteria (Supplementary Information 2.1) now provide more documentation on the selection/inclusion/exclusion criteria for the literature relevant in this study. In fact, we did not omit studies that report effects of moderate increases or natural variability in rainfall. Rather, we added a description of the magnitude of the considered rainfall event, variability or treatment in Table S3. It is indeed the case that almost all studies report effects of heavier-than-usual rainfall or substantial experimental increases. See additions in Supplementary Information 2.1, Figure S2 and Table S3.

7. on lines 116-118: Could you explain here or in the Methods by how much you increased the heavy rainfall and why you chose that amount and not another one? I.e. could you explain your reference scenario and “HR” scenario more clearly?

In the original manuscript we state the amount of the three heavy rainfall events in the methods section (previously L346-347, now L435-436). The baseline summer as well as winter precipitation is noted in Table 2. However, we did not explicitly state the amount of daily precipitation, which is described as the total summer/winter precipitation equally distributed over the amount of days warmer or colder than 0°C, respectively. We have now added this information as an extra column to Table 2 as well as added this information to the definition of the ref. and HR case.

See Tables 1 and 2 and L415-417: Rainfall is set to 60 mm and 300 mm for the dry and wet scenario, respectively, and covers the lower and upper range of observed average summer precipitation observed at the weather stations in Table 1.

8. on line 210 the authors mention the “uniform” increase of precipitation scenario a bit out of the blue, without introducing the reason of this scenario very well. Since the conclusions are important (soil response does not depend on rain frequency) I think it would be worth to better introduce this scenario with the other two (“HR” and “reference”).

We realize that the context of the “uniform” case was not well described. We have now improved the description of the relevance of this scenario in the discussion section to better tie it in. As the purpose of this scenario is a sensitivity analysis, we introduce this scenario in the discussion after the presentation of the main results from the HR and ref. cases

L210-221: Firstly, we investigate the sensitivity of magnitude and temporal distribution of increased summer rainfall by creating three additional scenarios. One with a uniform increase of overall summer rainfall by 50% to simulate for a general increase in summer precipitation as predicted by Bintanja et al. (²¹) (see Supplementary Information 3.2), and two scenarios accounting for even stronger rain events (50% stronger HR case) and less intense rain events (50% less intense HR case, see Supplementary Information 3.3). We find that generally under all tested conditions, the effect direction is unchanged, while the effect extent varies from up to 0.1°C warming in the dry cases and -0.15°C cooling in the wet cases for the uniform rainfall distribution (Fig. S4), to a warming of more than 0.3°C in the dry climate cases and a similar cooling of -0.15°C in the wet cases for the 50% more intense heavy rainfall event simulation (Fig. S5). This suggests that independent of the magnitude and temporal distribution, increased summer precipitation in the form of uniformly increased summer rainfall as well as the occurrence of heavy rainfall events will generally warm the ground in continental climates, but might lead to a cooling of the ground in maritime climates. However, the magnitude of warming/cooling as well as delays in freeze-up depend on the intensity of the rainfall events, calling for accurate representation of these events in larger-scale models.

Definitions of topsoil and subsoil

9. The division of subsoil and topsoil is quite important in this study. I think it would be important to explain and define these concepts upon first mention in the Introduction section already (line 52).

We added information on how we categorized measured depths in the literature into top- and subsoil and added a reference to Table S3.

L59-61: If divergent effects at different depths are reported, the effect is further distinguished into topsoil and subsoil warming or cooling (here, topsoil: ≤ 10 cm and subsoil: > 10 cm). Where quantified, we determine top- and subsoil depths in the literature based on reported measurement depths (see Table S3).

10. As I understand it, the authors define “topsoil” as the soil column exactly at the depth of 25% length of the maximum active layer thickness length (ALT) and “subsoil” as the soil exactly at depth of 75% of maximum ALT? Please describe this more clearly and it would be helpful if the authors added

We added a sentence in the methods describing that the depths of topsoil and subsoil are defined as the closest mesh element to the relative corresponding depths

L383-384: In the post-processing of model results, topsoil and subsoil depths are defined as the closest mesh element to the depth 25% and 75% of the maximum annual thaw depth.

10.1) an explanation of why these thresholds were chosen and why they are adequate, if possible, citing previous research or previous analyses of the literature data at hand

The thresholds were chosen to report both the effect in the subsurface close to the surface (but not at the surface) and the processes close to the permafrost table (but not in/at the permafrost/permafrost table) to investigate how different the responses of heavy summer rainfall are to different soil depths. To our knowledge, there is no standardized way of comparing different depths in the top- vs. subsoil with regard to active layer thicknesses in the literature. However, we now elaborate more on the choice of the division in the main text according to the comments below.

10.2) a discussion of the repercussions of that choice – i.e., does the choice of this relative definition versus fixed absolute boundaries for topsoil and subsoil affect how the results need to be interpreted?

Ground temperatures are based on modeled temperatures for each individual cell of the mesh. Naturally, comparing different absolute depths, especially in deeper soil layers, will lead to vastly different results if a cell located in the permafrost is compared to a cell within the active layer. Differences might be less severe in layers closer to the surface. The only way of consistently comparing topsoil and subsoil is hence to use relative values for both depths. To add more transparency to our approach, we have included a figure that shows all relative depth layers in 20% steps (see comment #10.3 below). In response to the comment we have further added a Figure (Fig. R1 below) in this letter that shows a selection of absolute depths visualized in the same way as the relative depths in Fig. R2 (below). It shows that a meaningful comparison between cold and warm climate cases in deeper depths is not possible using absolute depths (e.g., 70cm is located below the permafrost table in the cold climate cases).

10.3) the magnitude of the variance of topsoil and subsoil absolute depths among study sites and modelled climate scenarios (cf. line 311)? Do they vary a lot? Could you provide an argument or evidence that this does not matter for your conclusions?

We have added a table stating the absolute depths of the topsoil, subsoil, and maximum active layer thickness in Table S5. We have also added a Figure to the SI showing additional depth layers (surface, 20%, 40%, 60%, 80%, permafrost (table), alongside the chosen top- and subsoil depth of 25% and 75%, respectively) to provide additional information (Fig. S17 in the SI and Fig 21 below). This shows that our choice of relative depths does not influence our overall conclusions. We now refer to this information in the methods section.

L389-390: Temperature differences at additional depth layers (surface to ALT, in 20% steps) are shown in Fig. S17.

Figure R1: Temperatures differences between the ref. and HR case from the first heavy rain event until the end of the thawing season in the four climate scenarios (a-d) at different **absolute** depths in the subsurface from 0cm (surface) to 70cm as well as our chosen depths for topsoil (25%) and subsoil (75%). Daily values are averaged over a 7-day window.

Figure R2 (Fig. S17): Temperatures differences between the ref. and HR case from the first heavy rain event until the end of the thawing season in the four climate scenarios (a-d) at different **relative** depths in relation to the ALT from 0% (surface) to 100% (permafrost table) as well as our chosen depths for topsoil (25%) and subsoil (75%). Daily values are averaged over a 7-day window.

Literature review description

Due to the limited amount of peer-reviewed studies on the topic of this manuscript, it was not informative to conduct a systematic review as per definition (as systematic searching yielded very few useful results, see supplementary information S2.1). We therefore called it a “literature overview” in the original manuscript, which causes the lack of description as well as meta-analysis pointed out by the reviewer. However, we agree that the methodology is lacking in transparency and reproducibility and have added information to the methods as well as the Supplementary Information to address this.

Based on Cook (2019), we now describe our review method as a scoping review, in which we used a combination of systematic database search and purposive literature screening (based on snowball search, Google Scholar alerts) to identify relevant studies. Details on the procedure can be found in the detailed comments below, the new Supplementary Information section 2 and in the revised version of the manuscript.

11. The literature review is not very well described, neither in the Results nor in the Methods section.

- In the Methods section, it would increase the clarity of the study, if you described

11.1) Where (on which databases) you searched for the literature

11.2) Provide the exact key-word string and date of the search

11.3) Provide the criteria for inclusion or exclusion of literature

11.4) Ideally, it would be possible to describe the initial number of studies and studies excluded at several stages of the screening and reading process (cf. Liberati et al. 2009)

11.5) Did studies you collected report on soil temperature effects to heavy rainfall events (i.e. as you define it as extreme rainfall?) or just rainfall in general? Did you consider warming measured in the same year or at the end of the season or even warming in the next year?

Liberati, A. et al. The PRISMA statement for reporting systematic reviews and metaanalyses of studies that evaluate health care interventions: explanation and elaboration. *J. Clin. Epidemiol.* 62, e1-34 (2009).

We thank the reviewer for highlighting this need for clarification and reproducibility as well as the useful paper on literature review methodologies. We have now improved the literature review by adopting a formalization of search criteria as suggested by the paper; we have also written an extended Supplementary Methods section (section 2) in which we carefully explain the steps and our rationale for our review. In summary, we have:

1. clarified the type of review we have performed (a “scoping review”, following definitions of Cook, 2019)
2. added the databases searched, date accessed and exact key string we used
3. added results and strategies for additional search strategies apart from database search (snowballing and Google Scholar alerts)
4. added PICOSicos criteria for inclusion of studies into the literature overview, Table S2.
5. we also clarify the selection procedures for the studies used in the statistical evaluation of reported effects against site climatology (Fig. 2) using extended PICOS criteria for inclusion of studies into statistical analysis of reported soil thermal effects against climatological variables
6. elaborated on the rationale and description of our statistical analysis both in the main text and in the Supplementary Information
7. summarized our review approach in a PRISMA diagram, including the amount of studies initially identified and retained for various stages in the selection procedure.
8. added descriptions and rationale of several exceptions and additional decisions we have made in the discussion of the synthesized literature (Supplementary Information section 2.1.4)
9. explained that for the literature overview presented in Table S2, we also describe and consider next-year warming/cooling effects. However, for consistency in the statistical evaluation, we limit the selection to studies reported effects on active layer thickness, thaw depth or soil temperature within the same summer for the selection of studies evaluated in Figure 2.

Re-evaluation and formalization of our criteria did lead to some minor adjustments of studies identified in the literature and studies selected for the statistical analysis that were previously inconsistent. Based on a careful re-check of our search strategies (database search, snowballing, google scholar alerts), we have added a study by Lopez et al. 2010 and a study by Zhang et al. 2021, which previously have not been part of the literature overview, but have high relevance for our research question. We further omitted the study by Iijima et al. (2010)'s study from the statistical analysis of association between site climatology and reported effects of heavy rainfall, since Iijima et al. (2010) only report next-summer effects under combined heavy rain- and snowfall. This would lead to potential confounding effects and inconsistency in the timing of reported effect (since other studies generally report same-summer effects). We also take into account the reviewer's recommendation of not spatially aggregating studies, but use Moran's I statistic as an evaluation of potential issues with spatial dependence of residuals. Despite these changes, the identified pattern of reported response (warming/cooling/both/no effect) against continentality presented in (updated) Figure 2 remains largely the same. The observed pattern against winter temperature however was no longer evident from either the PCA (Fig. 2a) or Wilcoxon rank sum tests.

The resulting changes are too many to cite here separately and are discussed in more detail under comments following this overarching comment.

See new addition in Supplementary Information 2.1 and 2.2, describing the entire workflow from literature search to Table S2 (literature overview) to Fig. 2 in more detail.

See changes in description in main manuscript Methods, Line 292-314

See updated Results and description thereof in main manuscript, Line 45-46, 53-61, 76-93, Figure 2.

12. In the Results section, literature review findings tend to be described in a qualitative way, and it would add clarity and rigour to your study, if you could be a bit more concrete and quantitative in your description and add relevant references to literature or Figures/Tables in your study. For example:

Due to the heterogeneity in result and data presentation and description in the identified literature, it was not possible to conduct a strictly quantitative meta-analysis in this study. Instead, we adopted a vote counting procedure (Bushman & Wang, 2009), based on a rule set described in the Supplementary Information 2.2.1 and main methods. We have also added references to the categorization of the studies in the text according to the comments below.

13. I think it would add to the clarity of the text if you could provide references for claims on lines 31-33
Added a reference to Subin et al. (2013).

14. Also, on lines 37-38, you describe that "in the literature, opposing effects of heavy summer rainfall on ground temperatures have been reported in different studies...", however there are no references. Here it would be good to add such references to the corresponding studies.

Here, we allude to the 22 studies identified in the literature overview as stated in the following sentence. Adding all 22 references at this point feels disruptive for the flow of reading. Hence, we have not added the references here, but they are mentioned several times in the following section based on the comments below with all studies cited at least once in L66-68 where we explain how studies are categorized as cooling/warming/both/none.

15. Lines 49-51: "Effects of heavy rainfall on permafrost has seen increasing attention in scientific literature in recent years, with an increasing number of studies that add a modeling analysis to their methodology. A detailed overview of the identified studies is presented in Table S2." – Here it was not clear to me what it means to "add a modeling analysis"?

We modified the sentence slightly for clarity.

L50-52: Effects of heavy rainfall on permafrost has seen increasing attention in scientific literature in recent years, with increasingly more studies using a combined field- and modeling approach to investigate the effect.

16. Lines 57-60: Here some clarity needs to be added: how did you define topsoil and subsoil? Did you take your own definition or according to the definition of each corresponding study? Also, it would be adequate to give a reference to the claim “As subsoil temperatures are decisive for thaw depth development”.

We added a definition for top- and subsoil for the literature based on reported observational depths in the literature (Table S3 and comment #9). We did not add a reference to “As subsoil temperatures are decisive for thaw depth development”, but rewrote the sentence for clarity.

L59-62: If divergent effects at different depths are reported, the effect is further distinguished into topsoil and subsoil warming or cooling (here, topsoil: ≤ 10 cm and subsoil: > 10 cm, see Table S3). As soil temperatures close to the thaw front (subsoil) are indicative of thaw depth development we further classify a study as "warming", if subsoil warming is observed and "cooling" if the subsoil experiences cooling.

17. For example, when you describe the studies on line 57 ff., it would be nice to reference the corresponding studies which are alluded to. For example, I was not sure if the “two studies that address effect of wetter soil conditions through soil moisture,…” are now in addition to or if they are part of the 20 studies found in the literature review? Which ones are they in Table S2?

We agree that it is useful to add references to those statements in the text and added them here and for the following comments as suggested.

L63-64: We also consider two studies that address the effect of wetter soil conditions through soil moisture, which is not necessarily always coupled to precipitation^{32;33}.

18. Lines 60-62: could you please add the references of the studies you allude to here?

Added a reference to the study with “no effect”.

L62-63: Furthermore, one study did not see a temperature response to heavy summer rainfall (“no effect”)³¹

19. Similarly, on line 69, you describe “In general, studies that report thermal effects over multiple soil depths find warming of subsoils and cooling of the topsoil” – It would be important to add a reference to one of your tables or analyses so that this claim can be backed by a bit more rigor.

Added a reference to Table S3.

L71-72: In general, studies that report thermal effects over multiple soil depths find warming of subsoils and cooling of the topsoil (see Table S3).

20. The same on line 70: “Not all the studies report temperature response but rather report changes in ALT..” which studies are alluded to here? Please be a bit more precise.

Added references to the studies alluded to here based on Table S4.

L72-73: Not all studies report temperature response but rather report changes in ALT^{15;32;32;40} or surface heat fluxes⁴⁴ to describe the effect of heavy summer rainfall.

21. Lines 80-81: “We observed...higher continentality...in sites that showed warming effects of heavy rainfall

events” – Did all the studies report effects of heavy rainfall or just increased rainfall? By how much did rainfall increase in these cases?

We added a column to Table S3 to list the added magnitude of precipitation for the studies that report these quantities and mention this detail in the main text..

See Table S3 and L90-93: Although the current body of literature is limited and variable in methodology and rainfall event magnitudes, this suggests that warming effects of heavy rainfall events may mostly be observed in sites with continental climates and, accordingly, warmer summer and colder winters, whereas cooling effects may be observed in sites with maritime climates.

22. In Table S2: It seems that at least some observational studies are experiments – could you add this nuance? We have added this information as part of Table S3 together with the effect extent, reported depths and magnitude of added precipitation in a new column “study design”.

See Table S3

Literature review analysis

23. Several aspects of the literature review analysis, especially the data aggregations appear a bit arbitrary, are vaguely described and not well justified. If you could add some justifications and explanations of the decisions you made, that would make the analysis more convincing.

We have made significant efforts to more clearly describe and justify the decision making process. We have incorporated a more elaborate description and workflow for the identification of studies and how we assigned them to different response classes based on a rule set (which is indeed best described as a form of vote counting). We have also added explanations of how we further selected sites for statistical evaluation of relations between the outcome of the vote-counting procedure and site climatology.

See new addition in Supplementary Information 2.2 (PICOS criteria for inclusion of studies into statistical analysis, vote counting procedure and statistical approach).

See description of this workflow in the main manuscript Methods, Line 292-314

See updated Results and description thereof in main manuscript, Line 45-46, 53-61, 76-93, Figure 2.

24. On line 72-73 you describe that rainfall effects on soil temperatures can strongly depend on the duration and depth of monitoring of the soil temperatures. Have you considered this when statistically comparing the results of the literature review studies? If not, please explain why.

We judged that with the limited number of studies left ($n = 14$) for statistical comparison of climate variables across response classes, addition of covariates such as observation depth or duration would leave too few degrees of freedom. Instead, we have standardized the duration of observation to a large degree by only using the response class (warming, cooling, no effect, both effects) observed in the same summer as the heavy rainfall event or treatment, see new Supplementary methods. This did lead to omission of some studies from the statistical comparison (Iijima et al., 2010), and a change in response class for another study (Zhirkov et al., 2022, based on first-year effect only). Hence, we expect that we have reasonably minimized potential confounding effects based on the timing of observation relative to the rainfall treatment or event.

Depth of observation is another potential confounding factor to account for. We tried to mitigate this risk by using direct observations of active layer depth and thaw depth where-ever possible, to report changes at the thaw front. However, studies reporting cooling effects relatively often only reported shallower soil temperature changes (e.g. Sullivan et al., 2008 with changes at 10cm depth, Christiansen et al., 2012, with observations at

5cm depth). Hence the reviewer is right here in pointing out this potential confounding effect. However, with the highly limited amount of data points, adding such methodology-related covariates would yield little basis for a useful comparison. Instead, we now discuss this limitation of the dataset related to observation depths in the second discussion paragraph:

L186-195: Our results (Table S2, Table S3, Fig. 3) indicate an influence of depth of observation or changes to the soil thermal regime, with cooling predominating in shallow soil layers even when thaw depths or deeper soil temperatures indicate warming. Hence, studies reporting effects of heavy rainfall events on the soil thermal regime may show limited comparability across observation methods (e.g. active layer thickness, instantaneous thaw depths, soil temperatures or heat fluxes) and observation depths. While our evaluation of reported rainfall effects against site climatology (Fig. 2) showed a balanced distribution of experimental (e.g. irrigation) and observational studies across sites where warming and cooling effects were reported (Table S2), a relatively higher proportion of studies reporting shallower soil temperature changes among sites that report cooling (Table S2) may have contributed to observed patterns. However, our model simulations independently show a similar pattern of subsoil warming and active layer deepening under climatic conditions representative of continental sites, versus cooling and no effects on active layer thickness in climates representative for maritime regions.

Additionally, we point out potential remaining inconsistencies in L90-93: Although the current body of literature is limited and variable in methodology and rainfall event magnitudes, this suggests that warming effects of heavy rainfall events may mostly be observed in sites with continental climates and, accordingly, warmer summer and colder winters, whereas cooling effects may be observed in sites with maritime climates.

See new Supplementary Information 2.2.2.

See additions of observation depths in Table S3

See addition in main text discussion cited above, in Lines 186-195

25. The literature review covers a relatively small set of studies with different methods and time intervals of investigation. It would add to the clarity of the manuscript if information about measuring depth, magnitude of precipitation increases and timescales of the literature studies would be indicated in Table S4, for example. We agree with the reviewer that this is valuable information. While the timescale of each study is already part of Table S2 and measuring depths is part of Table S3, we added information on the magnitude of precipitation increase and the study design (where available) to Table S3 according to comments #21 and #22 above. Ideally, we would have added all this information in a single table, but due to space limitations we had to split the information up into 2 tables (Tables S2 and S3).

26. On line 76, It was not easy for me to understand why you analyzed climatological data only from 16 of the 20 studies? Please clarify.

The rationale for doing so was that we wanted to compare a site's climatology to the reported effect of rainfall. Several of the studies identified in literature and discussed in Table S2 and results, are larger scale studies or model studies that are not tied to a specific location and climatology, or do not report field-observed effects. Moreover, we wanted to avoid comparing first- and second-year (or later) effects, which may differ in direction or magnitude (see comment above). This is why we now describe the extended PICOS criteria for inclusion of sites into the statistical evaluation in specific detail in Supplementary Information 2, and reasons for exclusion in the PRISMA diagram, and we hope we have sufficiently clarified and justified our choice here.

See Supplementary Information 2.2

See clarification in methods in main text, lines 316-329

See added statement in results, L76-80: To reduce methodological differences among studies, we further narrowed down our selection to studies that report field- measured changes in active layer thickness, thaw depths or soil temperatures in spatially explicit locations in the same summer as the heavy rainfall event or experimental treatment (see Supplementary Methods 2.2). For the remaining 14 study sites, we find that sites for which warming effects of heavy rainfall were reported are generally situated in more continental regions compared to sites for which cooling effects were reported.

27. I think the type of analysis conducted with the literature data is more similar to a vote-counting approach rather than a meta-analysis, which would require effect sizes extracted from the collected studies (cf. Koricheva et al. 2014 and Gurevitch et al. 2018), hence, consider rewriting.

We agree, thank you for bringing this to our attention. We had no consistent reporting of effect sizes to base any formal meta-analysis on and hence resorted to reporting the general direction of the effect. We have added an extended description of the rule set on which we based the vote counting procedure, with reference to a study on meta-analysis and vote counting.

See Supplementary Information 2.2.2 and changes in Methods in lines 306-314: We used a vote-counting procedure⁶¹ to subdivide the identified studies into response classes, where we distinguished studies that report warming effects, cooling effects, divergent (both) effects or no effect of heavy rainfall, on either active layer thickness, thaw depths or soil temperatures. We assigned reported effects to response classes based on a rule set defined in Supplementary Information 2.2.1, where increases or decreases in active layer thickness, thaw depth or ground temperatures are considered "warming" and "cooling" effects, respectively. In cases where authors reported divergent effects (both warming and cooling) based on depth of soil temperature observation (e.g. cooling of topsoils but warming of subsoils^{17;42;43}), the effect reported at the deepest available soil depth was used, as this was assumed to be most representative of processes at the thawing front.

28. Lines 249-251: I think it is not adequate to simply exclude literature review studies that show no clear cooling or warming effect. Please provide a convincing reason to exclude this data from your analysis, if possible with reference.

We excluded these studies because there were few studies that reported such effects. Hence, we ended up with too few observations (< 5) in these categories to make solid comparisons of climate data across different response groups. But we understand that even so it was not justified to leave these studies out of the PCA analysis and Figure 2. We now include the "no effect" and "both effect" sites in all plots in Figure 2 and in the PCA analysis (however, the "both effects" studies were no longer retained based on updating of the extended PICOS criteria for inclusion into the statistical analysis, and for one study (Zhirkov et al., 2021), we changed "both" to "warming" to represent only the first-summer effect of their continued irrigation treatment, for consistency with the other retained studies). However, we still only report significance of contrasts among those response groups with sufficient observations ($n > 5$, warming and cooling response), using a non-parametric test to account for the fact that we still had a highly limited sample size. Hence, they are only excluded from the Wilcoxon rank-sum test, but included visually in all Fig. 2 and in the presented literature overview (Table S2) and discussion thereof.

See changes in Figure 2 and associated text (L292-305).

29. Similarly, on lines 263-264 you describe that results from sites less than 100 km apart were "averaged", even though averaging in this context means (as I understand) ascribing the most frequent value (i.e. in the case of three studies where two showed a cooling [Zhang and Zhou 2021] and one showed a warming [Li 2019], you ascribed a "cooling" effect [lines 266-267]). I have two questions: 100 km seems a large distance to me that not necessarily causes a spatial pseudo replication: on what basis did you decide to average sites less than 100 km apart? Could you add a reference or chain of arguments that supports this distance threshold? Additionally, I think it is not adequate to "average" three studies to a cooling effect, when in fact one of three studies did

show a warming effect? As in the case above (Lines 249 ff.), it would be more adequate to keep the original studies and their original results in the literature analysis.

The reviewer is right in pointing out that this is an arbitrary cut-off, which we identified based on a distance matrix among the sites' locations. With a limited body of evidence and high heterogeneity in tundra environments on very different spatial scales, we doubt whether it is possible to set a single, justified distance threshold a priori. As an alternative, we include all individual studies and report the Moran's I statistic of spatial association of residuals of the reported Wilcoxon rank-sum test, to ensure that spatial dependency of residuals does not incapacitate the reported p-values of Wilcoxon rank-sum tests. The main finding (contrast in continentality between warming and cooling sites) showed no indication of spatial dependency in the residuals (p value of observed vs. expected Moran's I > 0.05).

The reviewer will notice that due to stricter adherence to PICOS criteria and de-aggregating the previously merged studies, the outcomes in Fig. 2 have changed. The overall observed pattern however has remained the same.

We provide elaborate description of the site selection procedure and statistical analysis of the sites used in Figure 2 in Supplementary Information 2.2.2 - 2.2.4

See changes in reported statistics in Fig. 2.

30. Lines 252-254: Why did you take the monthly temperatures for the time 1959-2021 to calculate mean seasonal and annual temperature and precipitation and not a 30-year window closer to the temporal extent of the studies in the literature, e.g. 1990-2022? Additionally, on lines 255-256: Why did you choose again a different time (of 1981-2010) for the continentality index?

We see the point and agree, we now adopt a time frame of 1991-2020 for all climatological variables and explain that this is chosen to adapt to the time interval commonly used to characterize climate and representative of recent conditions (representative of the period under investigation in the studies we use).

See changes in Supplementary Information 2.2.3, and L330-336 in Methods.

Non-climatic factors

31. As you also describe in your study, not only climatic factors but also local characteristics of the landscape, vegetation, permafrost, and soil properties can affect the ground temperature response to increased rainfall. Is there a reason why you did not include such local soil type (e.g. water retention capabilities, organic layer thickness), permafrost type (e.g. ice-content) or vegetation type into the PCA of the literature review analysis (e.g. described on lines 74-79) and subsequent modelling analysis? The results would be more convincing if you could provide an argument why you did not (need) to consider soil type, permafrost ice-content or vegetation types for the main part of your analysis.

We understand the concern raised in this comment. We also agree that soil properties are important for understanding the comprehensive effect of heavy summer rainfall on ground temperatures. However, accurate information on appropriate spatial scales is sparse and highly heterogeneous on larger scales and most studies do not quantify soil physical properties or vegetation characteristics. We decided to focus on climatic factors in this study that can serve as a basis for future work on more regional scales, rather than addressing the full range of variables that can potentially affect the effect of heavy summer rainfall. Adding soil properties at this scale is not feasible to include in this study, and may result in high-dimension, low-sample-size issues with the currently limited body of evidence.

We explicitly frame the study as focused on climatological contrasts in heavy-rainfall response of permafrost soil thermal regime in the final paragraph of the introduction

L36-40: Here, we hypothesize that the effect varies with prevailing climatic conditions and collect relevant literature published on the effect of heavy summer rainfall on the active layer throughout different permafrost landscapes ranging from high Arctic over sub Arctic to Alpine permafrost regions. In order to understand the role of increasing rainfall and heavy rainfall events, we summarize and synthesize those findings, and categorize the effects into distinct processes that cause soils to react differently to heavy summer rainfall.

However, we improved the description of non-climatic factors and qualitatively discuss how they may potentially influence the observed effect

L238-250: Besides climatic factors, the landscape can influence the response of the active layer thermal regime by changing the path of water flow. Topography, hydraulic conductivity, and geomorphic features such as polygonal tundra can cause differences in moisture distribution and hence change how ground temperatures react to heavy summer rainfall for example through surface and subsurface runoff in sloped terrain, ponding of water in e.g. low-centered polygons, and runoff through trough networks in polygonal tundra landscapes^{8;27;55}. Our main results have shown that differences in soil moisture affect the response to heavy rainfall events. Soil moisture further depends on soil properties, water retention capacity, permafrost ice content, thermokarst development, vegetation, snow depth, etc., which thus may have an impact on how heavy rainfall will affect the ground thermal regime. How fast water can infiltrate, vegetation interception, how much water the soil can hold, and how it affects heat conduction and advection can have highly non-linear effects on the overall response to heavy summer rainfall^{10;14;56}. To address the sensitivity of our model results to differences in soil composition, and without changing any of the physical soil properties individually, we used the model to investigate the importance of the organic layer thickness to address some of the potential important influences mentioned above. The organic layer has a significant influence on water redistribution and thermal insulation of the soil and can hence alter the effect of heavy summer rainfall.

32. Also, could you add a description of how you identified the factors important to consider in your study: i.e. did you identify the climatic predictors from the collected literature? And which variables exactly did you test? For example, in line 84 you describe: “No significant contrasts in other climate variables were observed...” Could you provide a description of these other tested variables and the method how these were identified? We have improved the description of the statistical analysis and how we derived site specific data from the literature overview. The use of climatological factors stems from our own hypothesis that impacts of heavy rainfall on permafrost soils may vary across climatological regions (this hypothesis is now added explicitly at the end of the introduction, see previous comment). Climatic predictors are not provided by the studies identified in the literature, but by ERA5 data at the (approximate) location of the study site. We have split ERA5 data into seasonal statistics and tested whether seasonal climate statistics differed among response classes (warming, cooling, no effect, both effects). To inform our selection of comparisons to make, we first ran a PCA on site seasonal climate statistics and color these by response group (as explained in Supplementary Information and Methods section). Since the PCA showed some indication of alignment of continentality index and winter precipitation with the response classes (Fig. 2a), we now only statistically test these two seasonal climate statistics, and disregard other statistics.

We improved the description of the process in the main text

L82-90: To identify climatological contrasts that may explain variability in the soil thermal response across the current body of evidence, we performed a principal component analysis (PCA). The result indicates that the climatology of these sites mainly varied in terms of continentality and total summer and winter precipitation

(Fig. 2a, black arrows). We observed a pattern of higher continentality (and associated higher summer temperature and lower winter temperature) and lower winter precipitation in sites that showed warming effects of heavy rainfall events, as opposed to lower continentality and higher winter precipitation for sites that showed cooling effects (Fig. 2a). Wilcoxon rank sum tests confirm that sites for which warming effects are reported have significantly higher continentality indices ($p = 0.035$, $n = 14$, Fig. 2b,c) but indicated no significant difference in winter precipitation ($p = 0.731$, $n = 14$) compared to sites that showed cooling effects of heavy rainfall events.

.. and in the Supplementary Information, section 2.2.3, L922-934:

This resulting dataset of 14 sites with observed rainfall effect, coordinates and climate data was used to assess patterns among reported effects on the soil thermal regime (warming, cooling, both or no effect) and site climatological data (total precipitation and mean temperature for winter, spring, summer and fall, and CCI). We then assessed whether different responses were associated with contrasts in site climatology using a combination of exploratory (PCA) and inferential (Wilcoxon rank sum test) statistics. First, we ran a PCA on site climatological data (averaged temperatures and seasonal total precipitation per season and CCI) and visually assessed patterns of reported effects of heavy rainfall on the soil thermal regime against the first two principal components using a biplot. Since the biplot showed a strong alignment of the “cooling” and “warming” sites with CCI and winter precipitation (Fig. 2a), we tested whether CCI and winter precipitation differed significantly among sites that were assigned to different response classes during the vote-counting procedure (warming: $n = 7$, cooling: $n = 6$, none: $n = 1$, both: $n = 0$). We used non-parametric tests to account for relatively small sample sizes. Due to the low number of observations in the “both” and “no-effect” classes, we only tested for differences among the “warming” and “cooling” classes, using a Wilcoxon rank sum test and visualized results using boxplots. We assessed potential violation of spatially independent residuals using Moran’s I. We set our significance criterium at $\alpha = 0.05$.

And similar changes in the main text Methods, L316-329: We hypothesized that observed heterogeneity in the response of active layer temperatures to heavy rainfall events in current literature may be related to climatological contrasts. We therefore tested whether observed responses reported in various studies, resulting from the vote-counting procedure, were related to climatological contrasts among sites. To reduce variability in reported effects due to differences in study design and time span of observation, we implemented a second round of screening of the identified studies. For statistical analysis of association between reported effects of the selected studies and local climatic data, we narrowed the data selection to those studies that report field-measured data of soil thermal dynamics (soil temperatures, thaw depth or active layer thickness), within the same year as experimental treatment or naturally occurring heavy rainfall, and at spatially explicit locations. For consistency, we disregard studies reporting effects of only soil moisture increases^{33;43}, only next-season effects⁴¹ or only modeling results^{16;29;30;38;44} (see additional PICOS criteria for statistical evaluation in Supplementary Information 2.2). We acknowledge that this approach does not resolve all methodological differences among studies, and that differences in rainfall event or treatment magnitudes, differences among experimental and monitoring studies and use of different metrics (e.g. thaw depth and soil temperatures at variable depth) may influence our results. Still, we expect that the identified patterns provide a preliminary insight into potential contrasts in rainfall response of permafrost across climatological contrasts.

33. I think it is important to discuss additional, non-climatic factors affecting the soil thermal response to increased precipitation, such as on line 207 ff.

We have added and modified the paragraph on non-climatic factors not investigated in this study and how they potentially influence the results of this study as mentioned in our response to comment #31.

See comment #31.

Temporal extent of interest

34. I think it would greatly improve the clarity of the manuscript if the temporal extent (a season, a year or several years?) and resolution (daily, monthly, yearly?) of the collected literature and the modelling study was clearly described, if possible, in the Introduction or Methods section. As I understand, the modelling experiment for example is focused on within-season changes of the soil thermal regime due to heavy rainfall events, but no lagged or multiyear effects? Please clarify

We have added clarification on the temporal extent within the literature identification process and selection criteria for the statistical analysis in the revised manuscript (Tables S2 and S3) (please see our responses to comments #11, #12, and #23-30) . Wherever possible, we focus on within-season effects. We also improved the description of the modeling study analysis (see L127-128), in which, for the main results, we also focus on within-season effects. In addition, we have expanded our model analysis to include and discuss potential multi-rainfall years and lag-effect (see comment #49)

L126-128: The temperature difference is described as HR case temperature minus ref. case temperature in the topsoil (25% of the maximum ALT) and the subsoil (75% of the maximum ALT) on a daily time-step during the thawing season of the same year in which heavy rainfall was applied

Modeling analysis: selection of scenario and parameter values

35. I think the model experimental setup with the four different climate scenarios are well conducted and informative. However, there are some decisions on modelling scenarios and data processing that I find not very clearly explained. Specifically:

36. Line 210: "We also assessed the impact of heavy rainfall events compared to a uniform increase of precipitation by 50%." : this comes a bit out of the blue and it is not clear when and why you did that. It would be nice to better describe that in the Methods and Results sections.

We added a sentence in the methods to better clarify the intention behind this scenario and improved the description of the relevance of this scenario in both the methods and the discussion sections of the main text.

Methods:

L442-448: In order to account for the model sensitivity towards varying intensities in rainfall magnitude and temporal distribution, we also created scenarios that address equally/uniformly distributed increased summer rainfall (see Supplementary Information 3.2) and varying magnitudes of heavy summer rainfall events (see Supplementary Information 3.3). In the uniform scenario, summer rainfall is generally increased by 50% and therefore covers the other end-member along a heavy-rain-event to general-precipitation-increase gradient. In the scenarios with varying magnitude in rainfall events, we simulated 50% stronger heavy rainfall events as well as 50% less intense heavy rainfall events.

For the amendments to the Discussion, please see response to comment #8.

37. Lines 316 ff.: Why did you choose in-situ weather station data for a small selection of sites instead of calculating a more representative pan-Arctic average & range from ERA5? What were the variables that you retrieved from the stations as input to the ATS? I think it would increase the clarity of the study if you could add these variables to Table 1?

We decided to base our climate scenarios on weather station data instead of ERA5 because the focus of this study is heavy/extreme rainfall, which is generally underrepresented in the ERA5 dataset (e.g., Lavers et al, 2022). For the definition of the heavy rainfall scenarios according to the Weibull recurrence interval it was therefore crucial to capture the full range of heavy rainfall events based on observations. To improve for clarity, we added the information that we retrieved from weather station data to create the climate scenarios in Table 1, as suggested.

see Table 1

38. Lines 323 ff.: How did you define what is “representative” of cold and warm climates? Based on quantile or PCA-like analysis of the in-situ station data? Or based on other, literature derived criteria? It would be nice if you could add a more precise explanation here.

The cold and warm climates are defined by the in-situ station data and their respective location in continental/maritime climates. We have added information on average temperature, annual temperature amplitude, and summer rainfall sums to Table 1, which form the basis for the values shown in Table 2. We have also included this information in the text to improve for clarity.

L403-405: A warm climate represents a warm summer, but relatively cold winter (continental climate, e.g., Samoylov Island, see Table 1), while a cold climate is represented by a relatively cold summer, but mild winter (maritime climate, e.g., Zackenberg, see Table 1).

39. Line 330-333: How did you determine “low” and “high” precipitation scenarios? Was this decision based on predictions from previous research? Or based on quantile-analysis from Arctic-weather station data? Cf. also lines 340-342: Please explain why you chose exactly “a 100-year recurrence interval” precipitation event? Why did you simulate a 100-year recurrence interval event to happen 3 times in a row in June, July and August? 60 and 300 mm are indeed representative of the lower and upper end of observed average yearly precipitation at the weather stations listed in Table 1. To simulate future heavy rainfall events, we chose a 100-year recurrence interval as it provides a sound statistical basis for an “extreme” event. However, it is based on past weather observations, which likely leads to an underestimation when compared to actual future heavy rainfall. We decided to not use any climate model output to determine heavy rainfall events as the climate models are useful to determine overall trends in temperature and precipitation, but might not be able to simulate events such as we intended to simulate in this study. We have added additional information on the choice of 60 and 300 mm used for the “dry” and “wet” precipitation scenario.

L415-418: Total summer rainfall is set to 60 mm and 300 mm for the dry and wet scenario, respectively, and covers the lower and upper range of observed average summer precipitation observed at the weather stations in Table 1. Table 2 lists the key values used to describe the sine curve as well as precipitation sums and daily values.

40. Table 2: I would replace “MAAT” with “Tavg” and “AMP” with “Tamp” to be consistent with equation 2) Changed the table headers accordingly

41. Lines 335-336: How did you determine the parameters (minimum of six and maximum of 227 Wm⁻²? Are these parameters based on previous analyses or averages for the latitudes of interest? Please clarify. It would also be clearer to describe the exact equation instead of “similar to Eq. 2”

Our generic climates are intended to be representative for high Arctic as well as high mountain permafrost areas. However, due to the range of latitudes, a common, realistic incoming shortwave radiation distribution does not exist. Hence, we decided to represent a range from Arctic winter conditions (minimum of 6Wm⁻²) to subarctic summer conditions (227Wm⁻²), excluding any local variations through cloud cover. We have also now added an equation (Eq. 3) for shortwave radiation with the corresponding terms in the methods.

L421-424:

$$G(\text{doy}) = G_{\text{avg}} + \frac{G_{\text{amp}}}{2} \frac{\sin(2\pi*(\text{DOY} + C))}{365} \quad (3)$$

with $G(\text{doy})$ being the daily incoming shortwave radiation, G_{avg} being the average solar radiation (here: 116 W m⁻²), and G_{amp} representing the annual amplitude in solar radiation (here: 221 W m⁻²), resulting in an annual

minimum of 6 and a maximum of 227 W m^{-2} , which covers a range of Arctic winter insulation to sub-Arctic summer insulation.

42. Lines 337-338: “These values are within range of representative values in permafrost landscape climates and help to isolate the effect of only temperature and precipitation on the ground thermal regime.” – I think that is a valid argumentation – however could you provide a reference to what is representative for permafrost landscapes?

We have added a reference to the weather data observations in Table 1, which now also contains information about, relative humidity, and wind speed.

see Table 1 and L425-427: These values are within the range of representative values commonly observed in permafrost landscape climates (such as at the weather stations listed in Table 1) and, by keeping them constant, help to isolate the effect of only temperature and precipitation on the ground thermal regime.

43. Line 351-352: The implementation and results of this “relative” increase in precipitation scenario are not well explained, could you be a bit more elaborate on that? Why did you choose a 50% increase in precipitation and not more or less?

50% increase in summer precipitation is the upper range of what can be expected by the end of the century. in Bintanja et al. 2020, it is stated that Arctic precipitation is likely to increase by up to 40% over the first century with some areas experiencing an increase up to 50% (see Fig. 4b). We added a reference to the statement in the Supplementary Information.

L966-968: To assess the influence of a more uniform increase in precipitation, we performed a set of simulations in which total summer precipitation is increased by 50% representing the upper boundary of estimated overall future summer precipitation increase in the Arctic ²¹.

44. Generally, could you add in your discussion an elaboration on how your model assumptions affect the interpretation of your results? Are “real-world” effects likely to be over- or underestimated? Or do the parameter settings of your model not influence your findings?

Due to the physics-based nature of the model, thermal and hydraulic processes in the active layer are resolved individually and represent real-world effects explicitly. We obtain representative soil thermal and hydraulic parameters based on literature values because we aim for a generic representation of permafrost landscapes. The model conditions are realistic in this regard because the soil hydraulic and thermal properties used are relevant for permafrost soils. However, it is not possible to fully accommodate all soil configurations as these vary greatly, with variations in subsurface heterogeneity and not least due to non-linearity of water retention characteristics; covering this fully would go far beyond scope and intentions of our study. However, we have added more information and an assessment of the implications of changing the parameters in Supplementary Information 1.

L775-779: In our study we focus on generalizations of climatic conditions combined with generalizations of soil types, considering representation of organic and mineral soil textures with thermal and hydraulic parameters based on literature values. For cases where site-specific data is available, soil parameters can be calibrated against measurements to represent local site conditions. In our study however, we focus on larger-scale climatological contrasts. We do not aim to accurately resolve site-specific conditions, but rather obtain a general idea of the system behavior for class-type representations of organic and mineral soils.

Sensitivity analyses

45. I was wondering on what basis the precipitation scenarios (cf. Table 2) were chosen and how much the results would change if one would apply a smaller or larger amount of heavy rainfall. Did you explore in that direction? A sensitivity analysis with different magnitudes of heavy rainfall scenarios could shed light on the uncertainty and range of warming/cooling estimates.

As mentioned above (e.g. in response to comments #7 and #37), baseline precipitation is based on the observational weather data in Table 1. Extreme events are based on the 100 year recurrence interval for heavy summer rainfall. We agree that there is a lack of testing for the sensitivity of the added amount of precipitation. Therefore, we have conducted additional sensitivity analysis by modifying the heavy rain events to account for 50% heavier rainfall events as well as for 50% less intense heavy rainfall events (New Fig. S5 in the SI and Fig. R3 below). We have included this analysis in Supplementary information 3.3 together with the uniform rainfall distribution as they both refer to the amount of heavy rainfall and added a point of discussion in the discussion section of the main text.

See Supplementary Information section 3.3

and L210-221: Firstly, we investigate the sensitivity of magnitude and temporal distribution of increased summer rainfall by creating three additional scenarios. One with a uniform increase of overall summer rainfall by 50% to simulate for a general increase in summer precipitation as predicted by Bintanja et al. ⁽²¹⁾ (see Supplementary Information 3.2), and two scenarios accounting for even stronger rain events (50% stronger HR case) and less intense rain events (50% less intense HR case, see Supplementary Information 3.3). We find that generally under all tested conditions, the effect direction is unchanged, while the effect extent varies from up to 0.1°C warming in the dry cases and -0.15°C cooling in the wet cases for the uniform rainfall distribution (Fig. S4), to a warming of more than 0.3°C in the dry climate cases and a similar cooling of -0.15°C in the wet cases for the 50% more intense heavy rainfall event simulation (Fig. S5). This suggests that independent of the magnitude and temporal distribution, increased summer precipitation in the form of uniformly increased summer rainfall as well as the occurrence of heavy rainfall events will generally warm the ground in continental climates, but might lead to a cooling of the ground in maritime climates. However, the magnitude of warming/cooling as well as delays in freeze-up depend on the intensity of the rainfall events, calling for accurate representation of these events in larger-scale models.

Figure R3 (Fig. S5): Sensitivity analysis on temperature difference between HR and reference case from the first heavy rain event until the end of the thawing season in the subsoil and the topsoil as well as thaw depth development for (a) a scenario with 50% less intense heavy rainfall events as compared to the original scenario and (b) a scenario with 50% more intense heavy rainfall events in comparison to the original scenario. Differences are displayed as HR case minus reference case. Daily values are smoothed over a 7-day window. Grey vertical bars indicate the timing of the simulated heavy rain events in each summer month (June, July, August).

46. Similarly, it would help to interpret the significance of soil thermal responses to heavy rainfall compared to

temperature and continentality. Therefore, one could compare the magnitude of heavy rain-induced differences in topsoil and subsoil temperatures to the general temperature differences among the four climate scenarios, similar as is done for ALT in Figure 3 c).

46.2 Such analyses could help to better interpret the magnitudes and assess the robustness of predicted soil thermal responses to heavy rainfall.

To better interpret temperature differences in relation to continentality, we added a plot showing the actual difference as suggested in the comment to the supplementary information (see Supplementary Figures, Fig. S7 in the revised manuscript, and Fig. R4 below) and added a reference to this in the main text.

L125-128: We find that the four climate scenarios yield distinct opposing effects in the soil temperature response under heavy summer rainfall conditions. The temperature difference is described as HR case temperature minus ref. case temperature in the topsoil (25% of the maximum ALT) and the subsoil (75% of the maximum ALT) on a daily time-step during the thawing season of the same year in which heavy rainfall was applied (for absolute temperatures see Fig. S7 in the Supplementary Information).

Figure R4 (Fig. S7): Absolute temperatures in the ref. and HR case from the first heavy rain event until the end of the thawing season in (a) the subsoil (75% ALT) and (b) the topsoil (25% ALT). Dashed and solid lines represent the ref. case temperatures in each climate scenario, shaded areas indicate the change from ref. to HR case temperature. Daily values are averaged over a 7-day window.

Discussion of mechanisms

48. I think that some topics deserve more attention in the Discussion section, in particular:

o Soil moisture vs. Precipitation: as you already describe on line 61, soil moisture is not necessarily coupled to precipitation. It would be nice if you could elaborate a bit on the precipitation-soil moisture relationship in your Methods and/or Discussion section and explain what assumptions you made about that relationship in your modeling and how that influences the interpretation of your results? E.g. on Lines 147 ff. you describe the dry soils gain a strong increase in thermal conductivity due to increased saturation caused by heavy rainfall – are the modelled conditions realistic and representative for the permafrost areas of the Northern Hemisphere?

We have made attempts to improve the discussion section according to this suggestion. The relationship between infiltrating water/precipitation and soil moisture is governed by the water retention characteristics of the soil as well as its permeability and porosity. The model accounts for this combined with a saturation-dependent effective thermal conductivity, by calculating the fraction of pore space occupied by ice, unfrozen water and air (details provided in Painter and Karra 2014). The model conditions are realistic in this regard because the soil hydraulic and thermal properties used are relevant for permafrost soils. We have extended our discussion on possible effects of the land surface on potential redistribution of water (soil moisture) and the consequences for the effect of heavy summer rainfall. We have also elaborated on possible consequences of different soil properties and moisture conditions and how it may be affected by rainfall events.

L238-246: Besides climatic factors, the landscape can influence the response of the active layer thermal regime by changing the path of water flow. Topography, hydraulic conductivity, and geomorphic features such as polygonal tundra can cause differences in moisture distribution and hence change how ground temperatures react to heavy summer rainfall for example through surface and subsurface runoff in sloped terrain, ponding of water in e.g. low-centered polygons, and runoff through trough networks in polygonal tundra landscapes^{8;27;55}. Our main results have shown that differences in soil moisture affect the response to heavy rainfall events. Soil moisture further depends on soil properties, water retention capacity, permafrost ice content, thermokarst development, vegetation, snow depth, etc., which thus may have an impact on how heavy rainfall will affect the ground thermal regime. How fast water can infiltrate, vegetation interception, how much water the soil can hold, and how it affects heat conduction and advection can have highly non-linear effects on the overall response to heavy summer rainfall^{10;14;56}.

49. Timescales: Your analysis mainly focuses on intra-seasonal, “immediate” effects of rainfall. However, as you describe, effects of rainfall on the soil thermal regime can differ according to the timescale of interest (cf. eg. lines 95-98). Hence, it would be nice to add a short discussion on immediate versus cumulative, multiyear effects of increased rainfall on soil thermal regimes and permafrost thaw.

We appreciate the suggestion and have implemented a “multiyear consecutive rainfall” analysis as well a brief analysis on the lag-effects as this has also been requested by reviewer #3 (see comment #1). Specifically, we created one scenario in which a first year with three heavy rain events in summer is followed by 10 years of baseline rainfall as well as a scenario in which the heavy rainfall events are applied for four consecutive summers and then again followed by 10 years of baseline rainfall. We briefly discuss these results in terms of the intensification/inversion of the heavy rainfall effect in the SI and refer to it in the discussion in the main text.

L227-237: We conducted a brief multi-year model analysis, assuming a single year and four consecutive years with heavy summer rainfall events followed by ten years of baseline rainfall conditions. After a single heavy rainfall summer, we find lag effects and even an inversion of the warming effect in the years after a heavy rainfall summer (see Supplementary Information 4). Depending on the climate case, the rebound time to $<0.1^{\circ}\text{C}$ temperature difference between the HR and the ref. case in the subsoil can be between two (cold-wet) to eight (warm-wet) years of average conditions after a heavy rainfall summer (Fig. S6a). If several heavy rainfall summers occur in sequence, a dry climate might change towards a wet climate system response (Fig. S6b, cold-dry), and wet climate systems might experience an enhanced cooling as response to several heavy summer rainfall years. On the other hand, the rebound ($<0.1^{\circ}\text{C}$ temperature difference) in wet climates is significantly faster (three years in cold-wet and seven years in warm-wet) than in the dry climate cases, where the temperature difference after ten years is still larger than 1.5°C (Fig. S6b, cold-dry and warm-dry).

Figure X: Multiyear subsoil temperature differences over the summer period based on **(a)** a single year of heavy summer rainfall followed by ten years of baseline summer rainfall and **(b)** four consecutive years of heavy summer rainfall followed by ten years of baseline heavy rainfall. Values are shown for every other day and averaged over a 3-time-step-window.

Minor comments

Main text

50. Lines 71-72: “due to regional or local conditions,” I think this segment is not necessary for the argument at hand and more confusing, therefore I recommend removing it.

We removed this part of the sentence

51. Line 114: this claim could be supported by adding a reference to Figure 2d: “...central Siberian permafrost region (Figure 2d)”

We added a reference to Figure 2d

52. Lines 126-127: I don't exactly understand the meaning of this sentence, could you rephrase and clarify?
We have modified the sentence for clarity:

L133-134: While subsoil temperature responses show effects in opposite directions, topsoil temperature responses do not follow the same effect pattern.

53. Lines 236-237: what do you mean by "cross referencing" in this context? How did you do the key-word search and on which platform(s)? What was the exact key-string?

We changed the methods section based on the comments about the literature review methodology above (comment #11ff)

54. Lines 277: I would add the reference of Coon et al. 2019 here as well.

E.T. Coon, M. Berndt, A. Jan, D. Svyatsky, A.L. Atchley, E. Kikinzon, D.R. Harp, G. Manzini, E. Shelef, K. Lipnikov, R. Garimella, C. Xu, J.D. Moulton, S. Karra, S.L. Painter, E. Jafarov, and S. Molins. 2020. Advanced Terrestrial Simulator. U.S. Department of Energy, USA. Version 1.0. DOI

Added the reference accordingly

55. Lines 290-291: what is the size of the column mesh?

The column mesh discretization is variable with depth and has a high vertical resolution in the active layer (2 cm per cell) and decreases with depth (up to 2 m in the bottom of the domain). We have added the information about the cell thicknesses to the sentence describing the mesh discretization.

L364-366: The column mesh is discretized into several vertical cells, which are small near the surface of the domain (2cm thickness within the active layer) and increase in thickness with increasing soil depth (max. ~2m thickness at the bottom).

56. Lines 304-305: exchange "extreme precipitation" with "heavy rainfall" to be consistent with the terminology?

The terminology has been changed accordingly

Supplementary Material

57. Lines 630 ff: could you reference the data you allude to here? Is this from Table 1?

Added a reference to Table 1

58. Figure S3: "Differences are displayed as HR case.." – Did you mean "uniform precipitation increase case" instead?

Thank you for pointing this out, the figure caption was indeed incorrect and we have corrected it. The caption now reads:

Temperature difference in the uniformly increased precipitation scenario between the HR case and ref. case in (a) the subsoil, and (b) the topsoil, as well as thaw depth progression (c) Differences are displayed as HR case minus reference case. Daily values are averaged over a 7-day window

Reviewer #2

General comments:

This is a well-written and presented manuscript that will be of interest to the readership. Figures are easy to read and consistent. The results are noteworthy in providing an analysis to the potential role of increasing summer precipitation on permafrost thaw. The data analysis and methodology are well presented and provided in sufficient detail.

Overall the flow of the manuscript is good except I have a comment about the Conclusion section needing a rewrite (below).

I am not sure why the “hook” is the C feedback as there are many other reasons why permafrost thaw is bad for humans, ecosystems, infrastructure, hydrology, biogeochemical processes outside of C, hydrology, runoff, contaminants..... There is too much carbon cycle ramification and not enough of the myriad other reasons to study this. Why the yedoma map? It is not needed.

Fundamentally this paper is about permafrost thaw but there is no mention of “top down thaw” or “near-surface permafrost” which are critical components of the processes. There is also no mention of how/where increased precipitation can initiate or expand lateral thermokarst thaw degradation. Slope failure, lubrication/slumping along thaw fronts facilitated by water, and other processes should be at least mentioned. Some of this could also be in here instead of the yedoma maps and talk of C.

We would like to thank the Reviewer for taking the time to read and giving constructive feedback on our manuscript. We agree that a broader discussion on more potential consequences of amplified permafrost thaw through heavy summer rainfall will improve the overall relevance of the manuscript. One reason for the attention to carbon is that the regions that will most likely experience enhanced thaw through heavy summer rainfall coincide with large carbon- and ice-rich permafrost regions. Hence, this observation is central to our conclusion and therefore plays a significant role in this manuscript. However, we hope the scope has been broadened now, with help from the suggestions in comments.

Comments by line #:

1. 11-14: I recommend mentioning carbon briefly but there are also infrastructure, hydrology, agricultural, and ecosystem aspects of increasing top-down thaw of permafrost.

We have added additional threats of permafrost thaw to the landscape to the introduction as suggested by the reviewer.

L17-19: Furthermore, changes in active layer thickness can cause changes to the hydrological connectivity of the landscape⁹, to ecosystems and ecosystem services¹², as well as damage to infrastructure built on permanently frozen soils through soil subsidence and thermokarst^{13;14}.

2. 24: of the role

Corrected

3. 26: local increase where? The Arctic is quite expansive.

We included additional information on this in the sentence based on Bintanja et al. 2020, Figure 4b.

L25-26: Precipitation in the Arctic is expected to increase. Current estimates predict local increases of up to 40% (e.g., in the Canadian Arctic and Eastern Siberia) over the 21st century [...].

4. 90: into two groups. I recommend removing the #1 and #2 and say “into two groups.” First,, and then describe that. The say “The second mechanism is... and explain that. The numbers are confusing.

We removed the numbers accordingly

5. 89-100 Here I wonder if the role of added water in the basal soil leads to the creation of taliks or prevents/reduces the re-freezing of the ground in the following winter.

We did not see any mention of talik formation in the investigated literature studies. In our own model simulations, we don't see any formation of taliks as a consequence of increased summer rainfall either, but we do observe a delay in re-freezing in the autumn due to higher latent heat requirements. We added a sentence on this in the "driving physical mechanisms" section.

L171-172: Increased heat capacity in the active layer also leads to the observed delay in the onset of freezing (see Fig. 3c).

6. Figure 3. Maybe put the topsoil as a and the subsoil as b then keep thaw depth c. Makes for more of a "top-down" presentation.

We appreciate the suggestion and after reconsideration have not changed the order of the plots because we wish to maintain the focus on the subsoil temperature responses rather than topsoil responses

7. When it rains the temperature in a given area drops (compared for example to warm sunny conditions). Is this addressed in the model? If so, how/where? If not can you provide some sort of addressal in the text? In ATS rainwater is assigned the temperature of air temperature. Although it is right to assume that given prevailing weather conditions are likely to alter the temperature of rainwater, the temperature difference between air- and rain water temperature is overall small (Byres et al., 1949). We added this information about the model assumption in the Supplementary Information.

L786-787: In ATS, rain temperature is assumed to be equal to air temperature, which is a reasonable assumption⁷⁹ given the complex nature of meteorological conditions.

8. 192: were observed

Corrected

9. 206" without an organic layer

Corrected

10. 210-215: Any sense that heavier rainfall has less residence time due to rapid runoff and thus that rain water does not have as much impact? Is that process accounted in the model?

The process is indeed accounted for in the model. We have looked at the surface runoff output and did not see any runoff produced by the model, indicating that all of the water either infiltrates or evaporates. We have added a discussion point about potential surface runoff processes.

L239-242: Topography, hydraulic conductivity, and geomorphic features such as polygonal tundra can cause differences in moisture distribution and hence change how ground temperatures react to heavy summer rainfall for example through surface and subsurface runoff in sloped terrain, ponding of water in e.g. low-centered polygons, and runoff through trough networks in polygonal tundra landscapes^{8;27;55}

11. 213: "not very sensitive" is vague.

We have rewritten this paragraph according to suggestions by Reviewer #1. It now reads:

L210-221: Firstly, we investigate the sensitivity of magnitude and temporal distribution of increased summer rainfall by creating three additional scenarios. One with a uniform increase of overall summer rainfall by 50% to simulate for a general increase in summer precipitation as predicted by Bintanja et al. (21) (see Supplementary

Information 3.2), and two scenarios accounting for even stronger rain events (50% stronger HR case) and less intense rain events (50% less intense HR case, see Supplementary Information 3.3). We find that generally under all tested conditions, the effect direction is unchanged, while the effect extent varies from up to 0.1°C warming in the dry cases and -0.15°C cooling in the wet cases for the uniform rainfall distribution (Fig. S4), to a warming of more than 0.3°C in the dry climate cases and a similar cooling of -0.15°C in the wet cases for the 50% more intense heavy rainfall event simulation (Fig. S5). This suggests that independent of the magnitude and temporal distribution, increased summer precipitation in the form of uniformly increased summer rainfall as well as the occurrence of heavy rainfall events will generally warm the ground in continental climates, but might lead to a cooling of the ground in maritime climates. However, the magnitude of warming/cooling as well as delays in freeze-up depend on the intensity of the rainfall events, calling for accurate representation of these events in larger-scale models.

12. Conclusion:

At current this is merely a regurgitation of the Discussion main points and the abstract. I recommend moving some of the broader research questions here instead. For example, the roles of the timing of the precipitation, the rate, the organic matter, etc.- the various things that were not addressed. This way you could postulate/hypothesize/recommend these things be studied.

We added two sentences about the potential of future research to enhance our understanding about the various effects of heavy summer rainfall based on our results. Analysis of sensitivity simulations remains part of the discussion (precipitation timing/rate and organic layer thickness).

L272-276: The question of whether future heavy rainfall in summer will warm or cool the ground in permafrost landscapes is ambiguous and depends on a multitude of interactions. With our study we have furthered the understanding of the impacts of climate change on permafrost landscapes and identified potential gaps in the design of field experiments. Going forward, more standardized protocols and more field-based evidence is required to increase confidence in the various reasons for the diverging effects observed in the literature and their impacts on large-scale permafrost thaw.

13. 263-268: was the slope, vegetation, or soil type also summarized from the study sites? I realize this may be a lot of work and may not be in all studies but am curious.

The role of slope/rapid runoff is also something to consider at least by providing a sentence on this at the right location (perhaps in the main text).

Given different focuses of the different studies in the literature and the lack of standardized observational setups, we were not able to extract this information from the individual studies. However, we have added a sentence to the paragraph on other non-climatic factors that can affect temperature response through heavy rainfall.

L238-250: Besides climatic factors, the landscape can influence the response of the active layer thermal regime by changing the path of water flow. Topography, hydraulic conductivity, and geomorphic features such as polygonal tundra can cause differences in moisture distribution and hence change how ground temperatures react to heavy summer rainfall for example through surface and subsurface runoff in sloped terrain, ponding of water in e.g. low-centered polygons, and runoff through trough networks in polygonal tundra landscapes^{8;27;55}. Our main results have shown that differences in soil moisture affect the response to heavy rainfall events. Soil moisture further depends on soil properties, water retention capacity, permafrost ice content, thermokarst development, vegetation, snow depth, etc., which thus may have an impact on how heavy rainfall will affect the ground thermal regime. How fast water can infiltrate, vegetation interception, how much water the soil can hold, and how it affects heat conduction and advection can have highly non-linear effects on the overall response to heavy summer rainfall^{10;14;56}. To address the sensitivity of our model results to differences in soil composition, and without changing any of the physical soil properties individually, we used the model to

investigate the importance of the organic layer thickness to address some of the potential important influences mentioned above. The organic layer has a significant influence on water redistribution and thermal insulation of the soil and can hence alter the effect of heavy summer rainfall.

14. 284: “setpus”

Should this be “setups” or maybe it is a word I do not know.

Corrected

15. 297: spelling of “thermo”

Corrected

16. 310: scenarios

Corrected

17. 321: most active layer measurements are made later than August. This may not be a critical issue with the modeling but is worth noting.

It is true that the active layer is still developing sometimes up until late September. In our study, we focus on JJA temperatures as commonly referred to as the summer period. Magnussen et al. 2022 have also shown that September precipitation in Siberia may not affect the same-summer ground temperatures but rather affect freeze-up and next-summer ground temperatures. Additionally, our cold climate cases (cold-dry and cold-wet) are characterized by sub-zero air temperatures throughout most of September, while the warm climate cases (warm-dry and warm wet) show sub-zero temperatures only in the second half of September. This would lead to uneven comparisons and non-linear interactions between added summer precipitation and increased snowfall and hence complicate our analysis significantly. We have noted this in the methods section for transparency.

L440-442: Despite active layer development often reaching into September in various regions¹⁷, we did not add a September event due to potentially confounding interactions between rain- and snowfall due to low air temperatures during late summer/autumn.

Reviewer #3

General Comments:

How enhanced precipitation and global warming can help thaw permafrost is a topic of great interest. Through a meta-analysis of papers from the circumpolar region, the authors assess the current role of precipitation in increasing permafrost temperatures over a wide area and present the impact of the manifested effect in dry and continental climates. This is a new approach. In addition, the numerical model experiments have succeeded in presenting specific processes that show how precipitation can manifest temperature increases in the active layer. These suggestions can be highly appreciated as important basic information when considering various effects of enhanced precipitation due to climate change in the future, such as permafrost degradation, greenhouse gas fluxes, and ecological and hydrological changes. I basically recommend the paper be published, and I really have no fatal comments on the manuscript. It is clearly written and well explained, and the argument is easy to follow. However, I believe two major issues are missing from the content of this paper's discussion based on these two methods.

We would like to thank the reviewer for the constructive feedback and suggestions on how to improve the manuscript. We have made an effort to address the comments below by adding additional analysis to investigate multi-year analysis in terms of both, multi-rainfall years and lag-effects of a single heavy rainfall summer. We have also included additional information on the water budget, which is explicitly resolved for in our model.

1) The first point is that the authors evaluate extreme precipitation as a single-year event. The impact on soil moisture and temperature structure in the active layer in a single year would not be significant. The problem is strongly amplified when there is a series of multi-rainfall years. In the central Lena River basin, a series of high summer precipitation years from 2004 to 2008 resulted in the oversaturation of soil moisture in the active layer and widespread deepening and thawing of frozen soil in the active layer. The lack of mention of multi-year effects may be a significant deficiency. I am also concerned about the relationship with the subsequent winter. Late summer rains have been shown to have a cooling effect on deep soils, which may delay winter freezing and consequently contribute to warmer temperatures the following year.

We appreciate the suggestion to look at multi-rainfall years and potential subsequent winter as well as summer temperatures and have included additional analysis accordingly. We have included two new sets of simulations: (1) A single heavy summer rainfall event year followed by ten years of baseline rainfall conditions and (2) four consecutive years with heavy summer rainfall events followed by ten years of baseline rainfall conditions. We believe that these simulations can be informative, but may be interpreted with care, as changes in e.g., the onset of snowfall, maximum winter snow cover, and onset of rain can vastly change the ground temperature response during and after the winter. We added an extra section in the Supplementary Information 4 and briefly touched upon the multi-year effects in the discussion section of the main text.

See comment #49 by Reviewer 1, Supplementary Information section 4, and L227-237: We conducted a brief multi-year model analysis, assuming a single year and four consecutive years with heavy summer rainfall events followed by ten years of baseline rainfall conditions. After a single heavy rainfall summer, we find lag effects and even an inversion of the warming effect in the years after a heavy rainfall summer (see Supplementary Information 4). Depending on the climate case, the rebound time to $<0.1^{\circ}\text{C}$ temperature difference between the HR and the ref. case in the subsoil can be between two (cold-wet) to eight (warm-wet) years of average conditions after a heavy rainfall summer (Fig. S6a). If several heavy rainfall summers occur in sequence, a dry climate might change towards a wet climate system response (Fig. S6b, cold-dry), and wet climate systems might experience an enhanced cooling as response to several heavy summer rainfall years. On the other hand, the rebound ($<0.1^{\circ}\text{C}$ temperature difference) in wet climates is significantly faster (three years in cold-wet and

seven years in warm-wet) than in the dry climate cases, where the temperature difference after ten years is still larger than 1.5°C (Fig. S6b, cold-dry and warm-dry).

2) Second, although the numerical modeling experiment gave equal amounts of rainfall in July, August, and September, the effect of rainfall on the active layer would change significantly in the summer months due to the budget between precipitation and evapotranspiration. Rainfall in late summer (especially in August and September) may modify the frozen soil's hydrothermal environment. Suppose the assessment is made in the Arctic and subarctic regions. In that case, more attention should be paid to the relationship of the surface water budget between the different vegetation zones (grassland, boreal forest, and tundra).

The model used in this study is a physics-based numerical model, which uses the surface energy balance to calculate energy and mass fluxes. These also include bare ground evaporation represented by near-surface water availability. In fact, evaporative fluxes can indeed remove substantial amounts of water (see Fig. R5 below, not shown in the manuscript). However, ATS does not account for transpiration by plants with deeper roots (such as for example in the Yakutsk region) and hence may underestimate total evapotranspiration. We agree that this can be important as vegetation is vastly different between different permafrost landscapes. In the context of this study, we did not account for differences in vegetation. However, we added a sentence in the Supplementary Information describing the model parametrization and in the discussion section of the main text to address this potential shortcoming.

L252-255: Further, variability in vegetation may influence the rates of evapotranspiration, which affects the amount of water ultimately reaching the subsurface. For example, tundra shrub vegetation has been found to intercept 15-30% of incoming rainfall⁵⁷. While our model accounts for bare-ground evaporation, transpiration is not explicitly accounted for in its current configuration.

and L787-789: Water can leave the model domain via evaporation. This process only accounts for bare-ground evaporation and does not explicitly resolve transpiration by plants.

Figure R5: Daily absolute evaporative flux in the HR and the ref. case during the thawing season with sums over the same period. Yellow and blue bars indicate daily precipitation in the dry and wet climate cases. Daily values are averaged over a 7-day window.

Specific comments:

Figure 1: It may not affect much, but Sporadic colors are not distinguishable.

This Figure has been updated with more recently published data on permafrost extent and a different color scale according to Obu et al. 2018 (see Figure. 1 in the main text).

Table S2: The study area seems to be listed incorrectly. (E.g., Lena Delta -> central Yakutia in Iijima et al. 2010) The reviewer is right; it should be “Central Lena River Basin”, as the authors of this study refer to it. It is indeed situated in Central Yakutia. We changed the study area to “Central Lena River Basin”

see Table S2

References:

- Bintanja, R., van der Wiel, K., Van der Linden, E.C., Reusen, J., Bogerd, L., Krikken, F. and Selten, F.M., 2020. Strong future increases in Arctic precipitation variability linked to poleward moisture transport. *Science advances*, 6(7), p.eaax6869.
- Bushman, B.J. and Wang, M.C., 1994. Vote-counting procedures in meta-analysis. *The handbook of research synthesis*, 236, pp.193-213.
- Byers, H.R., Moses, H. and Harney, P.J., 1949. Measurement of rain temperature. *Journal of Atmospheric Sciences*, 6(1), pp.51-55.
- Gurevitch, J., Koricheva, J., Nakagawa, S., & Stewart, G. (2018). Meta-analysis and the science of research synthesis. *Nature*, 555(7695), 175-182
- Koricheva, J., & Gurevitch, J. (2014). Uses and misuses of meta-analysis in plant ecology. *Journal of Ecology*, 102(4), 828-844.
- Lavers, David A., Adrian Simmons, Freja Vamborg, and Mark J. Rodwell. "An evaluation of ERA5 precipitation for climate monitoring." *Quarterly Journal of the Royal Meteorological Society* 148, no. 748 (2022): 3152-3165.
- Liberati, A., Altman, D.G., Tetzlaff, J., Mulrow, C., Gøtzsche, P.C., Ioannidis, J.P., Clarke, M., Devereaux, P.J., Kleijnen, J. and Moher, D., 2009. The PRISMA statement for reporting systematic reviews and meta-analyses of studies that evaluate health care interventions: explanation and elaboration. *Annals of internal medicine*, 151(4), pp.W-65.
- Lopez C, M.L., Shiota, T., Iwahana, G., Koide, T., Maximov, T.C., Fukuda, M. and Saito, H., 2010. Effect of increased rainfall on water dynamics of larch (*Larix cajanderi*) forest in permafrost regions, Russia: an irrigation experiment. *Journal of forest research*, 15(6), pp.365-373.
- Obu, J., Westermann, S., Bartsch, A., Berdnikov, N., Christiansen, H.H., Dashtseren, A., Delaloye, R., Elberling, B., Etzelmüller, B., Kholodov, A. and Khomutov, A., 2019. Northern Hemisphere permafrost map based on TTOP modelling for 2000–2016 at 1 km² scale. *Earth-Science Reviews*, 193, pp.299-316.
- Painter, S.L. and Karra, S., 2014. Constitutive model for unfrozen water content in subfreezing unsaturated soils. *Vadose Zone Journal*, 13(4).
- Subin, Z.M., Koven, C.D., Riley, W.J., Torn, M.S., Lawrence, D.M. and Swenson, S.C., 2013. Effects of soil moisture on the responses of soil temperatures to climate change in cold regions. *Journal of climate*, 26(10), pp.3139-3158.
- Zhang, G., Nan, Z., Zhao, L., Liang, Y. and Cheng, G., 2021. Qinghai-Tibet Plateau wetting reduces permafrost thermal responses to climate warming. *Earth and Planetary Science Letters*, 562, p.116858.

REVIEWERS' COMMENTS

Reviewer #1 (Remarks to the Author):

General impression

I congratulate the authors for their excellent work in revising their manuscript and addressing all my concerns. Their revisions are very thorough and all-encompassing, and greatly improved the clarity and robustness of their important and timely study. Hence, I highly recommend their manuscript for publication in Nature Communications, after considering a few minor persisting issues I describe below.

Minor comments

(I worked on the manuscript version with track changes: 397172_1_related_ms_7467745_rrzb7y.pdf)

Line 3 : Since you did not do a meta-analysis, replace “first meta-analysis” with “literature review” or similar?

Line 35 : I think it could improve the clarity of the manuscript if you briefly introduced the mechanisms leading to opposing effects of heavy summer rainfall on ground temperatures; i.e. effects on soil heat capacity and soil thermal conductivity. Additionally, it could be worth to already introduce that effects of heavy rainfall could be different depending on soil depth (i.e. topsoil vs. subsoil effects).

Figure 2: It would improve the clarity of the Figure if you explained the acronyms in the PCA biplot (Figure 2a), as well as the timespan and source of the climate data that was used to calculate climate variables in Figure 2a,b and c.

Figure 3: It would improve the clarity of the Figure if you wrote the full name of the “HR” scenario in the Figure legend upon first mention. Also: There is the description of “Grey vertical bars (...)” – However, I don’t see grey vertical bars in the Figures? Please check this for all Figures of this kind.

Figure 4: Consider mentioning “thermal conductivity” more prominently (e.g. as the first word) in the Figure legend, to highlight the content of the Figure more clearly.

Figure 5: Consider mentioning “heat capacity” more prominently (e.g. as the first word) in the Figure legend, to highlight the content of the Figure more clearly.

Line 379: Consider revising: “Tmax and Tmin are the annual maximum and minimum (daily) air temperatures”

Line 450: Consider revising: Tavg instead of MAAT.

Line 470: I might have missed it, but I don’t understand the meaning of the “daily values” in Table 2, could you clarify that?

Table 1: It would improve the clarity of the table if you added the acronyms (in the header of the table) to the explanations in the legend of the table.

Table 2: Could you add a more detailed explanation of the variables in the header of the table? For example, it is not entirely clear to me, what “C” stands for? (e.g. explicitly refer to Equation 2?). Also, I did not really get what the daily rain and snow variables (mm d-1) mean and in which scenario they were applied? Why are daily rain values different among all four scenarios and not only between the dry and wet?

Reviewer #2 (Remarks to the Author):

I applaud the authors for their detailed response to the Reviewer's comments. I hope they agree the paper is clearer, more richly developed, and of broader interest/impact. All of my comments/edits were adequately addressed.

Reviewer #3 (Remarks to the Author):

Thank you for the authors' thorough response to the many comments from the reviewers. I am pleased that the authors have added many new sensitivity experiments in a short period and have been able to evaluate the impact of precipitation on permafrost thawing in a more multifaceted manner. I believe the authors have appropriately understood the intent of my comments and summarized their importance within the purview of this paper.

I can conclude that the manuscript is close to being accepted, but I noticed a few other things in the revised manuscript. I would appreciate the authors' response if any corrections can be made.

Do the x-axis tick marks on time series charts mean the beginning of each month (1 Aug for Aug)? If so, it would be easier to understand if the labels were in the middle of the ticks. Figures 4 and 5 (+similar figures in the supplement) do not have tick marks for July and September, so please add them.

In the model results, the maximum ALT is around mid-August, which seems much earlier than when the ground surface freeze begins. This result was a bit curious because I think that ALT usually deepens just before the onset of freezing. Does this mean the cooling from the lower boundary condition (-9°C) works? It does not need to be corrected, but I am just wondering.

P.6, L.119: Isn't the inland of eastern Siberia more continental than central Siberia?

P.10, L.204-205: What specific terrain or landscape do you envision where lateral heat transfer could occur?

P.12, L.274-276: As the authors point out, I highly agree that observation based on standardized protocols is substantially recommended. In which areas of climatic and environmental conditions do you think it would be effective to make observations?

Reviewer #1

General impression

I congratulate the authors for their excellent work in revising their manuscript and addressing all my concerns. Their revisions are very thorough and all-encompassing, and greatly improved the clarity and robustness of their important and timely study. Hence, I highly recommend their manuscript for publication in Nature Communications, after considering a few minor persisting issues I describe below.

We would like to thank the reviewer for their time to read through and comment again on our revised manuscript and greatly appreciate their feedback. We believe that the comments raised in this review indeed help to further improve the clarity of the manuscript and have addressed them below (line numbers refer to the line numbers in the revised manuscript).

Minor comments

(I worked on the manuscript version with track changes: 397172_1_related_ms_7467745_rrzb7y.pdf)

1. Line 3 : Since you did not do a meta-analysis, replace “first meta-analysis” with “literature review” or similar?

We have indeed missed this mention of the word meta-analysis and have changed it accordingly.

L3-4: Here, we provide the first literature review of studies reporting on effects of rainfall on ground temperatures in permafrost environments and [...]

2. Line 35 : I think it could improve the clarity of the manuscript if you briefly introduced the mechanisms leading to opposing effects of heavy summer rainfall on ground temperatures; i.e. effects on soil heat capacity and soil thermal conductivity. Additionally, it could be worth to already introduce that effects of heavy rainfall could be different depending on soil depth (i.e. topsoil vs. subsoil effects).

We added information about the mechanisms causing changes in temperature response to heavy rainfall in the literature and mention that these differences may also vary with depth.

L35-38: Opposing effects of heavy summer rainfall on ground temperatures have been reported in the literature, often hinting at rain-induced changes in the hydrothermal soil properties (such as thermal conductivity and heat capacity) or changes to surface heat fluxes and the surface energy balance. Additionally, opposing effects are not only found between different sites, but also between different observation depths within the soil profile.

3. Figure 2: It would improve the clarity of the Figure if you explained the acronyms in the PCA biplot (Figure 2a), as well as the timespan and source of the climate data that was used to calculate climate variables in Figure 2a,b and c.

We have added the information to the figure caption.

Caption of Fig. 2: Climatological contrasts among sites for which field-measured effects of heavy rainfall on the soil thermal regime are reported based on monitoring or experimental studies. (a) PCA

biplot, where arrows represent PC loadings of averaged temperature (Temp) and precipitation (Prec) in spring (Spr), summer (Sum), fall (Fall, labels not shown here due to low PC loadings) and winter (Win) as well as the Conrad's Continentality Index (CCI) for selected sites (Table S4). Points represent sites, colored by reported effect of heavy rainfall on the soil thermal regime. (b) Boxplot of CCI of selected sites, grouped and colored by warming or cooling as reported effect. Center lines represent the median, box limits represent upper and lower quartiles, whiskers represent 1.5 times the interquartile range and points represent individual data points. Moran's I indicates no significant spatial autocorrelation among residuals. (c) Map of reported effects from selected sites over map of Conrad's Continentality Index, with hyperoceanic (CCI < 20), oceanic (CCI < 50), subcontinental (CCI < 60), continental (CCI < 80), and hypercontinental (CCI > 80) classifications. All meteorological data used in (a) and used to calculate CCI in (b) and (c) is based on ERA5 reanalysis for the time span 1991–2020.

4. Figure 3: It would improve the clarity of the Figure if you wrote the full name of the "HR" scenario in the Figure legend upon first mention. Also: There is the description of "Grey vertical bars (...)" – However, I don't see grey vertical bars in the Figures? Please check this for all Figures of this kind. We have added the full name for the heavy rain (HR) case. Gray vertical bars in this figure and all other figures show the timing of the rainfall events and they show up in our version of the pdf (and see below). Is it possible that there is a problem with the rendering of the pdf so that they do not show up in the reviewer's version?

Caption of Fig. 3: Temperature difference between heavy rain (HR) and reference case from the first heavy rain event until the end of the thawing season in (a) the subsoil (75% ALT) and (b) the topsoil (25% ALT).

5. Figure 4: Consider mentioning “thermal conductivity” more prominently (e.g. as the first word) in the Figure legend, to highlight the content of the Figure more clearly.

We appreciate the suggestion and changed the figure caption accordingly:

Caption of Fig. 4: Thermal conductivity throughout the depth profiles of each scenario displayed as daily differences between the reference case and the heavy rain (HR) case (HR-ref. case).

6. Figure 5: Consider mentioning “heat capacity” more prominently (e.g. as the first word) in the Figure legend, to highlight the content of the Figure more clearly.

We appreciate the suggestion and changed the figure caption accordingly:

Caption of Fig. 5: Heat capacity throughout the depth profiles of each scenario displayed as daily differences between the reference case and the heavy rain (HR) case (HR-ref. case).

7. Line 379: Consider revising: “Tmax and Tmin are the annual maximum and minimum (daily) air temperatures”

These minimum and maximum values are indeed average annual minimum and maximum values as the CCI is calculated over the long term average temperatures. We have changed the sentence slightly:

L340-341: where CCI is the Conrad's continentality index, T_{max} and T_{min} are the average annual maximum and minimum air temperatures, respectively, [...]

8. Line 450: Consider revising: Tavg instead of MAAT.

We agree that Tavg is more consistent with the rest of the equations and table and changed it accordingly and deleted the abbreviation in the brackets:

L403-404: For each of the sites we then calculated the average annual air temperature , [...]

9. Line 470: I might have missed it, but I don't understand the meaning of the “daily values” in Table 2, could you clarify that?

We agree that “daily values” sounds confusing. We therefore changed it slightly:

L423-424: Table 2 lists the key values used to describe the sine curve as well as precipitation sums in mm and daily precipitation rate in mm d^{-1} .

10. Table 1: It would improve the clarity of the table if you added the acronyms (in the header of the table) to the explanations in the legend of the table.

We added the acronyms to the table caption and also added the information for the last two columns, which we have missed to mention in the caption before.

Caption of Table 1: Information about the sites that observational weather data has been retrieved from in order to create the synthetic climate scenarios including information about location, observation length, data source, long-term average temperature (T_{avg}) and annual temperature

amplitude(T_{amp}), long-term average summer precipitation sums (summer P), average relative humidity (RH), and average wind speed (wind). [...]

11. Table 2: Could you add a more detailed explanation of the variables in the header of the table? For example, it is not entirely clear to me, what “ C ” stands for? (e.g. explicitly refer to Equation 2?). Also, I did not really get what the daily rain and snow variables (mm d^{-1}) mean and in which scenario they were applied? Why are daily rain values different among all four scenarios and not only between the dry and wet?

We understand the confusion and added some clarification to this specific information in the text and the caption of Table 2.

L417-419: Due to differences in air temperature development between the cold and the warm scenarios, this leads daily precipitation values not only being different between the dry and wet scenario, but also between the cold and warm scenarios.

And caption of Table 2: Summary of the synthetic climate scenarios. Average annual temperature (T_{avg}), average annual temperature amplitude (T_{amp}), and the phase shift factor (C , see Eq. 2) are needed for the sine curve that describes temperature, rain- and snow sum represent the sum of summer and winter precipitation, respectively, rain (mm d^{-1}) and snow (mm d^{-1}) indicate the daily amount of rain- and snowfall, respectively. Daily precipitation amounts are different for all scenarios due to the difference in infiltration period (rain: air temperature $> 0^{\circ}\text{C}$, and snow: air temperature $\leq 0^{\circ}\text{C}$) caused by differences in air temperature in the warm and dry scenario. Both temperature and precipitation values shown here represent the upper and lower range of observed climatic conditions at weather stations listed in Table 1.

Reviewer #2

I applaud the authors for their detailed response to the Reviewer's comments. I hope they agree the paper is clearer, more richly developed, and of broader interest/impact. All of my comments/edits were adequately addressed.

We would like to thank the reviewer for their very positive feedback and agree that their feedback and the comments from the other reviewers have contributed to significantly improve the quality of this study.

Reviewer #3

Thank you for the authors' thorough response to the many comments from the reviewers. I am pleased that the authors have added many new sensitivity experiments in a short period and have been able to evaluate the impact of precipitation on permafrost thawing in a more multifaceted manner. I believe the authors have appropriately understood the intent of my comments and summarized their importance within the purview of this paper.

I can conclude that the manuscript is close to being accepted, but I noticed a few other things in the revised manuscript. I would appreciate the authors' response if any corrections can be made.

We would like to thank the reviewer for their feedback and for providing recommendations on improving the clarity of our manuscript and agree that after incorporating the changes below, the manuscript is now even clearer (line numbers refer to the line numbers in the revised manuscript).

1. Do the x-axis tick marks on time series charts mean the beginning of each month (1 Aug for Aug)? If so, it would be easier to understand if the labels were in the middle of the ticks. Figures 4 and 5 (+similar figures in the supplement) do not have tick marks for July and September, so please add them.

We have added ticks and labels for July and September in the 2D plots (Fig4, Fig5, and FigS13). We have also changed the tick labels in all time series figures to include the day of the month (e.g. 1 Aug, 1 Sep, etc.) to clarify the meaning of the x-axis tick marks.

2. In the model results, the maximum ALT is around mid-August, which seems much earlier than when the ground surface freeze begins. This result was a bit curious because I think that ALT usually deepens just before the onset of freezing. Does this mean the cooling from the lower boundary condition (-9°C) works? It does not need to be corrected, but I am just wondering.

The thaw depth development in the model depends on the forcing on the surface of the model domain. We force the model with a surface energy balance based on synthetic weather data. Hence, the active layer development is not strictly representative for a specific site but rather a generalization of specific climatic settings (continental vs. maritime climate).

The maximum active layer thickness varies slightly between climatic and wetness conditions; indeed for the cold-dry case, the maximum is reached around mid-August, however for the other cases it generally is maximum towards the end of August. This is somewhat clearer in the updated figures where we have included ticks for each month, as suggested in the comment #1.

3. P.6, L.119: Isn't the inland of eastern Siberia more continental than central Siberia?

We changed "central" to "eastern".

L120-121: The warm and dry scenario with cold winters is representative of most highly continental regions with high annual temperature amplitudes, such as the eastern Siberia permafrost region.

4. P.10, L.204-205: What specific terrain or landscape do you envision where lateral heat transfer could occur?

We added two examples from the literature (bogs and hillslopes) where landscape features may cause advective heat transport through groundwater flow.

L206-207: The role of lateral heat advection as a response to summer rainfall is still unclear, but may be of particular importance in permafrost landscapes with distinct topographical features like bogs or along slopes.

5. P.12, L.274-276: As the authors point out, I highly agree that observation based on standardized protocols is substantially recommended. In which areas of climatic and environmental conditions do you think it would be effective to make observations?

We added suggestions for different climatic conditions in which field observations would be helpful to improve our understanding of the effect of summer rainfall on ground temperatures. We have not added suggestions for different environmental conditions or landscapes as we cannot draw conclusions about these based on the findings of our study. However, we agree that this will be a very important next step in the process of increasing our understanding about the full range of impacts of climate change on permafrost degradation.

L276-278: Going forward, more standardized protocols and more field-based evidence across different climatic conditions such as continental and maritime regions is required to increase [...]